# TabUnite: Efficient Encoding Schemes for Flow and Diffusion Tabular Generative Models

## Abstract

Flow matching and diffusion generative models for tabular data face challenges in modeling heterogeneous feature interrelationships, especially in data with continuous and categorical input features. Capturing these interrelationships is crucial as it allows these models to understand complex patterns and dependencies in the underlying data. A promising option to address the challenge is to devise suitable encoding schemes for the input features before the generative modeling process. However, prior methods often rely on either suboptimal heuristics such as one-hot encoding of categorical features followed by separated modeling of categorical/continuous features, or latent space diffusion models. Instead, our proposed solution unifies the data space and jointly applies a single generative process across all the encodings, efficiently capturing heterogeneous feature interrelationships. Specifically, it employs encoding schemes such as PSK Encoding, Dictionary Encoding, and Analog Bits that effectively convert categorical features into continuous ones. Extensive experiments on datasets comprised of heterogeneous features demonstrate that our encoding schemes, combined with Flow Matching or Diffusion as our choice of generative model, significantly enhance model capabilities. Our TabUnite models help address data heterogeneity, achieving superior performance across a broad suite of datasets, baselines, and benchmarks while generating accurate, robust, and diverse tabular data.

## 1 Introduction

Tabular data are ubiquitous in data ecosystems in many sectors such as healthcare, finance, and insurance (Clore et al., 2014; Moro et al., 2012; Datta, 2020). These industries utilize tabular data generation for many practical purposes, including imputing missing values, reducing sparse data, and better handling of imbalanced datasets (Jolicoeur-Martineau et al., 2024; Onishi & Meguro, 2023; Sauber-Cole & Khoshgoftaar, 2022). However, a particular challenge inherent to tabular data that generative models face is feature heterogeneity (Liu et al., 2023). Specifically, accounting for feature heterogeneity is vital in flow matching and diffusion-based generative models, since they rely on continuous transformations of denoising score-matching, or invertible mappings between data, and latent spaces (Ho et al., 2020a; Lipman et al., 2022). Unlike homogeneous data modalities such as images or text, tabular data often contain mixed feature types, ranging from (dense) continuous features to (sparse) categorical features. More importantly, these tabular features, regardless of form, are intertwined contextually (Borisov et al., 2023). For example, the numerical salary of a person is correlated with their categorical age and education (Becker & Kohavi, 1996). Therefore capturing the interrelationships between tabular heterogeneous features is crucial for flow and diffusion tabular generative models to incorporate contextual knowledge for understanding complex patterns and dependencies in the underlying data.

A promising solution for the feature heterogeneity challenge is to devise suitable encoding schemes for pre-processing the input features before applying the generative model. However, existing methodologies often rely on (1) separate generative processes on discrete & continuous features which do not model their correlations properly, (2) suboptimal encoding heuristics, or (3) learned latent embedding which is parameter inefficient. For example, the one-hot encoding approach for categorical variables leads to sparse representations in high dimensions, where generative models are susceptible to underfitting (Krishnan et al., 2017; Poslavskaya & Korolev, 2023). On the other hand, creating a latent embedding space requires training an additional embedding model such as ResNet

(He et al., 2015), or a Transformer-based $\beta$-VAE (Higgins et al., 2017; Kingma & Welling, 2013; Zhang et al., 2023), and trained using e.g., self-supervised learning (Chen et al., 2020). Hence, the quality of latent space generative models also depends on the embedding model's capability to capture the underlying dependency structure of the tabular data. To summarize, proper pre-processing of heterogeneous features is crucial for high-quality tabular data generation, and poor encoding schemes for the data features can lead to information loss that cannot be recovered from the generative model.

Our goal is to generate high-quality synthetic tabular data using proficient categorical encoding schemes to unify the data space. This enables a single flow or diffusion model to be applied, capturing crucial heterogeneous feature interrelationships. In summary, our contributions are as follows:

1. We introduce two novel categorical encoding schemes, PSK Encoding, and Dictionary Encoding, as well as leverage Analog Bits (Chen et al., 2022) from the discrete image domain which seamlessly converts categorical variables into an efficient and compact continuous representation. By facilitating the model to generate data in a unified continuous space, we can "unite" the mixed features to capture heterogeneous feature interrelationships based on a single flow/diffusion model on continuous inputs. Empirically, under our encoding schemes, the model learns to accommodate the heterogeneity of tabular features.

2. We conduct a comprehensive review between Flow Matching (Lipman et al., 2022) and DDPM (Ho et al., 2020b) as our generative model. Our results showcase that combining our categorical encoding schemes with DDPM attains state-of-the-art results on most settings. Conversely, Flow Matching speeds up the sampling speed dramatically, saving time and computation power, while yielding competitive results to DDPMs. Consequently, we propose the following models: TabUnite(i2b)-Flow/DDPM, TabUnite(dic)-Flow/DDPM, and TabUnite(psk)-Flow/DDPM. These models achieve superior performances across a wide spectrum of tabular data generation baselines, datasets, and benchmarks. The architecture of our models is illustrated in Table 1. Note that we also introduce TabFlow, a Flow Matching/Discrete Flow Model (Campbell et al., 2024) that models heterogeneous features separately, as a baseline.

3. We curate a large-scale heterogeneous tabular dataset from the Census dataset (Meek et al., 2001) with over 80 mixed continuous/categorical features, and over 2.4 million samples. This benchmark is significantly more challenging for tabular generative models than existing benchmarks from public data repositories (Dua & Graff, 2017; Vanschoren et al., 2013) which often have $\leq$ 100k datapoints and $\leq$ 30 features. It better reflects the scalability of tabular generative models, where our empirical results justify the importance of good encoding schemes for heterogeneous features.

## 2 RELATED WORKS

**Generative Models in Tabular Data Generation.** The latest tabular data generation methods have made considerable progress compared to traditional methods such as Bayesian networks (Rabaey et al., 2024) and SMOTE (Chawla et al., 2002). CTGAN and TVAE (Xu et al., 2019) were two models based on the Generative Adversarial Network (Goodfellow et al., 2014) and Variational Autoencoder (Kingma & Welling, 2013) architectures respectively. These models were applied along with techniques such as conditional generation and mode-specific normalization to further learn column-wise correlation. Other works such as GReaT (Borisov et al., 2023) and GOGGLE (Liu et al., 2023) saw successes with the use of graph neural networks and autoregressive transformer architectures respectively in performing tabular data synthesis. Recently, Diffusion (Ho et al., 2020b) and Flow Matching (Lipman et al., 2022) provided new avenues for exploration within the tabular domain. This included STaSy (Kim et al., 2022), which employed a score-matching diffusion model paired with techniques such as self-paced learning and fine-tuning to stabilize the training process, and CoDi (Lee et al., 2023), which used separate diffusion schemes for categorical and numerical data along with interconditioning and contrastive learning to improve the synergy among different features. TabDDPM (Kotelnikov et al., 2023) presented a similar diffusion scheme compared to CoDi and showed that the simple concatenation of categorical and numerical data before and after denoising led to improvements in performance. The most recent work in this domain was TabSYN (Zhang et al., 2023), a latent diffusion model that transformed features into a unified embedding via a feature tokenizer before applying EDM diffusion (Karras et al., 2022) to generate synthetic data.

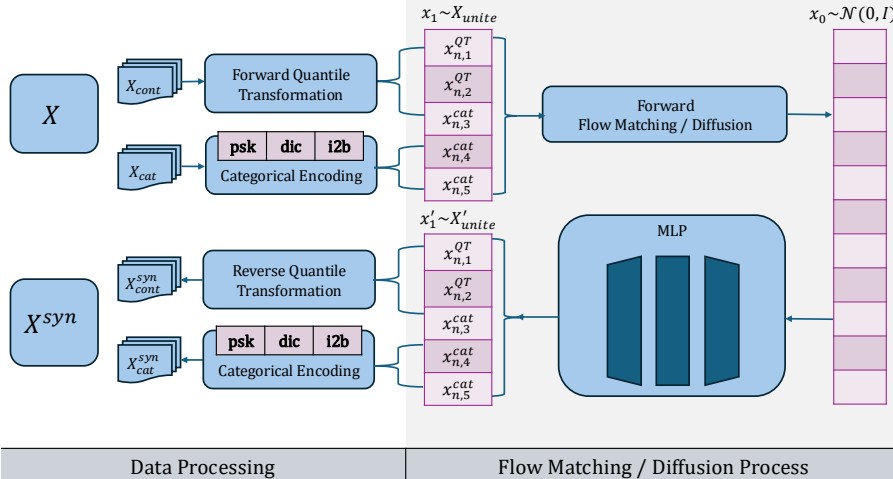

Figure 1: TabUnite(psk/dic/i2b)-Flow/DDPM Architecture. Continuous features $x^{cont}$ are encoded via a QuantileTransformer (Pedregosa et al., 2011). Categorical data $x^{cat}$ are encoded using Analog Bits or Dictionary Encoding methods. With an efficient continuous data space, we apply Conditional Flow Matching as our generative model where we ultimately synthesize samples. These samples are then mapped back to their original representation via their respective decoding schemes.

**Encoding Schemes.** CoDi (Lee et al., 2023) and TabDDPM (Kotelnikov et al., 2023) utilized a separated data space, where Gaussian Diffusion (Ho et al., 2020b) was performed on numerical columns and Multinomial Diffusion (Hoogeboom et al., 2021) was performed on categorical columns, with some additional techniques used to bind the two separate diffusion models. However, learning the cross-correlation among various features through separate methods was often less effective than conducting diffusion directly across a unified data space that included all features in the dataset. To achieve this, various encoding schemes were employed to process both categorical and numerical data so they occupy the same data space. One of the most widely used methods was one-hot encoding, which was used in both STaSy (Kim et al., 2022) and TabSYN (Zhang et al., 2023) that encoded categorical columns. One-hot encoding transformed categorical variables into a binary vector, where each category was populated with 0's with the exception of a single 1 that indicated the presence of a particular category. On top of one-hot encoding, TabSYN (Zhang et al., 2023) further used a column-wise feature tokenization technique that together transformed numerical and categorical features all into shared embeddings of the same length.

**Flow and Diffusion Generative Models.** Flow methods were introduced to the field of diffusion-based deep generative models as Probability Flow ODEs (Song et al., 2021), which, originally based on the concept of normalizing flows (Rezende & Mohamed, 2016), allowed for deterministic inference and exact likelihood evaluation. Compared to other diffusion-based methods such as score-matching (Song et al., 2021), DDPM (Ho et al., 2020b), and DDIM (Song et al., 2022a), flow-based models used continuous transformations defined by neural ODEs, to map samples from a simple distribution to a more complex target distribution. This allowed for efficient density estimation and generation of high-dimensional data. In the context of tabular data, Flow Matching was applied to gradient-boosted trees in place of neural networks to learn the vector field (Jolicoeur-Martineau et al., 2024).

## 3 TabUnite Models

Before diving into our methodology, we begin the section with preambles regarding a high-level overview of the tabular setting. Here a tabular dataset is characterized as $\mathbf{X} = \{\mathbf{x}_i\}_{i=1}^{N}$ with $N$ samples (rows), where a datapoint $\mathbf{x}_i \in \mathbb{R}^{D_{\text{cont}}} \times \mathbb{N}^{D_{\text{cat}}}$ comprises of $D_{\text{cont}}$ continuous features and $D_{\text{cat}}$ categorical features. We denote each $\mathbf{x}_i$ as $\mathbf{x}_i := [x_{i,1}^{\text{cont}}, \cdots, x_{i,D_{\text{cont}}}^{\text{cont}}, \cdots, x_{i,1}^{\text{cat}}, \cdots, x_{i,D_{\text{cat}}}^{\text{cat}}]$.

Our goal is to generate synthetic data samples, $\mathbf{x}^{syn}$, that mimic the quality of the real data, $\mathbf{X}$. To do so, we are required to learn a parameterized generative model known as $p_\theta(\mathbf{X})$, from which $\mathbf{x}^{syn}$ can be sampled. Prior to learning, extensive data pre-processing is required where categorical features are encoded into continuous features: $f(x^{cat})$, where $f$ denotes the encoder. Poor or sparse feature

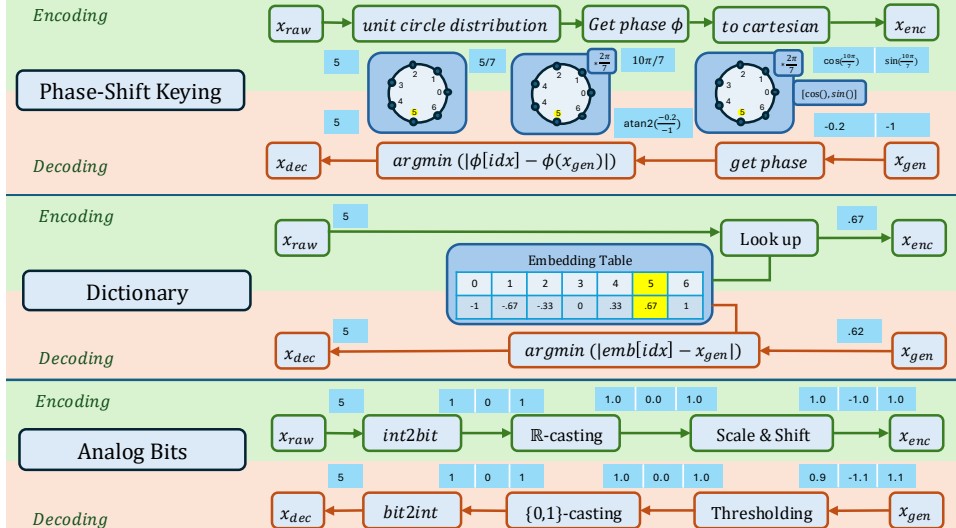

Figure 2: TabUnite Encoding Methods. We leverage PSK, Dictionary, and Analog Bits encoding to transform categorical features into a compact and efficient continuous representation before applying a single unified generative model to synthesize tabular data.

encoding of categorical features can hinder the model's ability to learn effectively. Therefore, we devise efficient and effective encoding schemes to address this issue.

## 3.1 ENCODING SCHEMES

We explore PSK, Dictionary, (naming conventions inspired by (Carson, 1922; Mairal et al., 2008)), and Analog Bits (Chen et al., 2022) to encode categorical features. An overview of our methods can be found in Figure 2 and Table 1. One-hot encoding typically lead to high-dimensional sparse vectors (Poslavskaya & Korolev, 2023) and causes underfitting when learning from it (Krishnan et al., 2017). In contrast to traditional one-hot categorical encoding, our encoding methods offer more efficient and dense representations while being able to incorporate crucial feature interrelationships.

Note that continuous features are encoded using the QuantileTransformer (Pedregosa et al., 2011) per TabSYN's and TabDDPM's methodology (Zhang et al., 2023; Kotelnikov et al., 2023). Additionally, the order in which the categories of a feature are assigned in our encoding schemes is based on lexicographic ordering for simplicity purposes. In the following sections, we consider an element of a categorical feature (single cell in a table), $x_{i,j}^{\text{cat}}$, that has $K$ unique categories: $x_{i,j}^{\text{cat}} \in \{0, \ldots, K-1\}$.

PSK ENCODING – TABUNITE(PSK)

PSK encodes categorical embeddings using a phase-based representation. Each category is assigned a unique phase angle in the complex plane, effectively transforming categorical variables into continuous, circular representations. The $K$ values will be evenly distributed as points on the unit circle, such that the $k$-th category is positioned at phase $\theta_k = \frac{2k\pi}{K}$. This phase value can easily be translated to cartesian coordinates in a complex domain, with a real component of $\cos(\theta_k)$, and an imaginary component of $\sin(\theta_k)$ as per Euler's Formula. The real and complex components are then concatenated to create our PSK-encoding in the form of:

$$ f_{psk}(x_{i,j}^{\text{cat}}) = \left[ \cos\left( \frac{2x_{i,j}^{\text{cat}}\pi}{K} \right), \sin\left( \frac{2x_{i,j}^{\text{cat}}\pi}{K} \right) \right] \tag{1} $$

For example, assume 4 categories denoted by the set $x_{i,j}^{\text{cat}} \in \{0, 1, 2, 3\}$. After PSK encoding, we obtain: $\{0, 1, 2, 3\} \rightarrow \{[1, 0], [0, 1], [-1, 0], [0, -1]\}$. These are the coordinates of 4 dots at 0, 90, 180, and 270 degrees of the unit circle. To perform decoding, the PSK encoded values are converted to their original categorical form by calculating the phase angle of the real and imaginary components using $atan2$, and then mapping this angle to the nearest category.

| Cat. Encoding Schemes | Cat. Dimensions | Examples |
|---|---|---|
| One-Hot | $\sum_{j=1}^{D_{\text{cat}}} K_{:,j}$ | $[0, 0, 1, 0] \to \text{Dim.} = 4$ |
| TabUnite(i2b) – Analog Bits | $\sum_{j=1}^{D_{\text{cat}}} \lceil \log_2(K_{:,j}) \rceil$ | $[0, 1] \to \text{Dim.} = 2$ |
| TabUnite(dic) – Dictionary | $D_{\text{cat}}$ | $[-0.33] \to \text{Dim.} = 1$ |
| TabUnite(psk) – PSK | $2 \cdot D_{\text{cat}}$ | $[0, 1] \to \text{Dim.} = 2$ |

Table 1: Comparison of Encoding Schemes. $K$ and $D_{\text{cat}}$ denote the number of unique categories per categorical feature and the total number of categorical features respectively. For the examples, we assume one categorical feature with four unique categories, and we encode the value $x_{i,j}^{\text{cat}} = 1$.

PSK encoding has an *equidistant representation* which ensures that all categories are treated equally in terms of their relative positions. This helps maintain a *compact* representation while avoiding the unintended bias that might arise from arbitrary ordinal encodings. Additionally, its circular continuity inherently captures the *cyclical nature* of certain categorical variables such as periodic events in a calendar and financial fiscal quarters.

DICTIONARY ENCODING – TABUNITE(DIC)

Dictionary encodes categorical features using a look-up embedding table function. This function encodes the categories to equally spaced real-valued representations within a range from $-1$ to $1$. Note that when a categorical feature contains more categories, the embedding may require a larger range to prevent the values from being too close to each other, which could hinder the model's ability to distinguish between categories. This can be addressed by tuning the range accordingly.

We begin by defining our one-dimensional Dictionary encoding function as:

$$f_{dic}(x_{i,j}^{\text{cat}}) = -1 + \frac{2x_{i,j}^{\text{cat}}}{K - 1} \tag{2}$$

For example, assume that there are 5 categories denoted by the set $x_{i,j}^{\text{cat}} \in \{0, 1, 2, 3, 4\}$. After one-dimensional Dictionary encoding, we obtain the following: $\{0, 1, 2, 3, 4\} \to \{-1, -0.5, 0, 0.5, 1\}$. This encoding ensures the preservation of the intrinsic order in ordinal data. In our experiments, we use the one-dimensional encoding setup described above. To perform decoding, the Euclidean pairwise distance between $x_{i,j}^{\text{syn}}$ and each of the $K$ categorical embeddings is calculated. The categorical value corresponding to the nearest embedding vector is selected.

Dictionary Encoding can be extended to $n$ dimensions to capture more nuanced patterns in complex datasets. We create an embedding matrix $\mathbf{M} \in \mathbb{R}^{K \times n}$ by first filling $\mathbf{M}$ with randomly sampled values from a standard normal distribution $\mathcal{N}(0, 1)$. Then normalize $\mathbf{M}$ by scaling the values of each column linearly between range $-1$ and $1$, using each column's minimum and maximum values.

Dictionary encoding preserves the *intrinsic ordering* among ordinal categorical data in the embedding space. This preservation of order is crucial for maintaining the *semantic relationships* between categories, especially where the *sequence* matters. By mapping categories to equally spaced values, it ensures that the relative distances between categories in the original data are reflected in the encoded representation. This compact format reduces dimensionality and improves model performance.

ANALOG BITS ENCODING – TABUNITE(I2B)

Analog Bits encode categorical features using a binary-based continuous representation. The encoding process involves two steps. First, we convert the categorical value to a binary representation where each category can be expressed using $\lceil \log_2(K) \rceil$ binary bits based on the number of categories. For example, a categorical feature with $K = 5$ categories is expressed using $\lceil \log_2(5) \rceil = 3$ bits that maps $x_{i,j}^{\text{cat}} \in \{0, 1, 2, 3, 4\}$ to $x_{i,j}^{\text{cat}} \in \{000, 001, 010, 100, 101\}$ respectively. Subsequently, each binary bit is cast into a real-valued representation, followed by a shift and scale formula.

$$f_{i2b}(x_{i,j}^{\text{cat}}) = (x_{i,j}^{\text{cat}} \cdot 2 - 1) \tag{3}$$

where $x_{i,j}^{\text{cat}} \in \{0, 1\}^{\lceil \log_2(K) \rceil}$.

This transformation shifts and scales the binary values from $\{0, 1\}$ to $\{-1.0, 1.0\}$. Thus, training and sampling of continuous-feature generative models (e.g., diffusion models) become computationally tractable. To decode, thresholding and rounding are applied to the generated continuous bits from the model to convert them back into binary form, which can be decoded trivially back into the original categorical values.

Analog Bits provide a dense representation which *reduces dimensionality*. It improves memory efficiency by requiring only $\lceil \log_2(K) \rceil$ dimensions per feature instead of $K$ in one-hot. This higher information density helps preserve heterogeneous feature relationships in a compact space.

**Overview.** In Figure 2, we consider an example categorical data point of $x_{i,j}^{cat} = 5$ with $K = 7$ categories where $x_{i,j}^{cat} \in \{0, 1, 2, 3, 4, 5, 6\}$. PSK's phase-based representation is obtained by mapping $x_{i,j}^{cat} = 5$ to $f_{psk}(x_{i,j}^{\text{cat}}) = \left[ cos(\frac{10\pi}{7}), sin(\frac{10\pi}{7}) \right]$. Dictionary creates a look-up embedding table and maps $x_{i,j}^{cat} = 5$ to $f_{dic}(x_{i,j}^{\text{cat}}) = .67$. Analog Bits encode $x_{i,j}^{cat} = 5$ using $\lceil \log_2(7) \rceil = 3$ bits, followed by casting into $\mathbb{R}$ then a scale and shift yielding: $f_{i2b}(x_{i,j}^{\text{cat}}) = [1.0, -1.0, 1.0]$.

Note that out-of-index (OOI) can occur due to the generative nature of models like Flow Matching and DDPM, which may produce continuous values that don't directly correspond to the original categorical encoding range. For PSK, OOI values are cast to the value of the 0-th index i.e., for 2 categories, we have $\{0, 1\} \to \{[1, 0], [0, 1]\}$. If we encounter an OOI category like 4, it would be cast to the value of the 0-th index. So: $4 \to 0 \to [1, 0]$. For Dictionary, OOI values are cast to the closest value in the embedding look-up table. For Analog Bits, OOI values are cast to the bit representation of the numerically largest value. Although casting ensures that all generated categorical values fall within the valid range, it introduces a bias. However, due to the generative capabilities of Flow Matching and DDPM, out-of-index values rarely occur as shown by the performance of TabUnite.

## 3.2 FLOW MATCHING AND DIFFUSION MODEL

After encoding our continuous and categorical columns, we are presented with a unified and continuous data space, $\mathbf{X}_{\text{i2b}} \in \mathbb{R}^{N \times \left( D_{\text{cont}} + \sum_{j=1}^{D_{\text{cat}}} \lceil \log_2(K_{:,j}) \rceil \right)}$, $\mathbf{X}_{\text{dic}} \in \mathbb{R}^{N \times (D_{\text{cont}} + D_{\text{cat}})}$, and $\mathbf{X}_{\text{psk}} \in \mathbb{R}^{N \times (D_{\text{cont}} + 2 \cdot D_{\text{cat}})}$. This enables us to directly model continuous state flow and diffusion models without requiring a discrete state space (multinomial diffusion/discrete flow) or re-formulation of the continuous flow/diffusion process (latent spaces). For convenience, we define $\mathbf{X}_{\text{unite}}$ to represent either $\mathbf{X}_{\text{i2b}}$, $\mathbf{X}_{\text{dic}}$ or $\mathbf{X}_{\text{psk}}$, depending on the encoding method used.

Subsequently, we examine Flow Matching (FM) (Lipman et al., 2022) and Denoising Diffusion Probabilistic Models (DDPMs) (Ho et al., 2020b) as our generative models where we apply our encoding methods prior to the modeling process. FM is a simulation-free framework for training continuous normalizing flow models (Chen et al., 2019) by replacing the stochastic diffusion process with a predefined probability path constructed with theories from optimal transport (McCann, 1997). On the other hand, DDPM learns a reverse denoising process by gradually transforming Gaussian noises into data samples.

Our encoding schemes, combined with FM and DDPM, yield our TabUnite models which are referred to as TabUnite(i2b)-Flow/DDPM, TabUnite(dic)-Flow/DDPM, and TabUnite(psk)-Flow/DDPM respectively. Please find additional details regarding Flow Matching and DDPM in Appendix A, B, Lipman et al. (2022), and Ho et al. (2020b). The following delineates the formulation for FM and DDPM in our setting.

Let $x$ denote a sample from the dataset $\mathbf{X}_{\text{unite}}$, i.e. $x \sim \mathbf{X}_{\text{unite}}$. For FM, We learn a vector field $v_t(x)$ to approximate the true vector field $u_t(x|x_1)$, yielding our objective function of the following:

$$L_{\text{FM}}(\theta) = \mathbb{E}_{q(x_1), p_t(x|x_1)} ||v_t(x) - u_t(x|x_1)||^2 \tag{4}$$

For DDPM, we learn a noise prediction network $\epsilon_\theta(x_t, t)$ to approximate the noise added during the forward process, yielding our objective function of the following:

$$L_{\text{DDPM}}(\theta) = \mathbb{E}_{t, \mathbf{x}_0, \epsilon_t} \|\epsilon_t - \epsilon_\theta(x_t, t)\|^2 \tag{5}$$

Relative to DDPM, FM synthesizes tabular data with a much higher sampling speed while also attaining a competitive generalization.

## 4 Experiments

We evaluate the performance of TabUnite(i2b)-Flow/DDPM (Analog Bits + FM/DDPM), TabUnite(dic)-Flow/DDPM (Dictionary encoding + FM/DDPM), and TabUnite(psk)-Flow/DDPM (PSK encoding + FM/DDPM) on a wide range of real-world and synthetic datasets, benchmarks, and compare the proposed models with a comprehensive number of baselines. Hyperparameters can be found in Appendix B.3. In the figures, we use TabUnite(psk) as our exemplars. Figures of other TabUnite methods can be found in the Appendix as we reference them in their respective sections.

**Datasets.** The datasets in our experiments are from the UCI Machine Learning Repository (Dua & Graff, 2017), L2X paper (Chen et al., 2018), and our own self-curated dataset, "Census Synthetic". The real-world UCI tabular datasets have previously been utilized in existing baselines. Next, we leverage synthetic toy datasets to prove the faithfulness of our model (Appendix C.1 and D.3). Lastly, we curate a dataset much larger than existing datasets in the number of samples (approx. 2.5 million samples) and include a large set of mixed features (40 and 41 categorical and continuous features each). The training/validation/testing sets are split into 80/10/10% apart from the Adult dataset which we adhere to its original documented splits. Full details of the datasets can be found in Appendix C.1.

**Baselines: Existing modeling approaches.** We compare our model against eight other existing methods for tabular generation. This includes CTGAN (Xu et al., 2019), TVAE (Xu et al., 2019), GOGGLE (Liu et al., 2023), GReaT (Borisov et al., 2023), TabDDPM (Kotelnikov et al., 2023), STaSy (Kim et al., 2022), CoDi (Lee et al., 2023), and, TabSYN (Zhang et al., 2023). SMOTE (Chawla et al., 2002), is also included as a base reference model. Results from CTGAN, TVAE, GOGGLE, GReaT, STaSy, and CoDi are taken from the TabSYN paper (Zhang et al., 2023). The main competitors to our model are TabSYN and TabDDPM since they are the best-performing models to date. Hence, we reproduce the results of TabSYN and TabDDPM per the recommended hyperparameters mentioned by the authors of their respective papers. More details in Appendix C.2.

**Ablations: Encoding schemes and generative models (Flow/Diffusion).** We conduct our ablation studies with respect to various encoding schemes and generative models. This assists us in proving the effectiveness of our encoding schemes (PSK, Dictionary, and Analog Bits) and highlights FM's (Lipman et al., 2022) competitive yet fast performance against DDPMs. Details in Appendix C.3.

**Benchmarks & Metrics.** We evaluate the generative performance on a broad suite of benchmarks from TabSYN (Zhang et al., 2023). We analyze the capabilities in *downstream tasks* such as machine learning efficiency (MLE), where we determine the AUC score for classification tasks and RMSE for regression tasks of XGBoost (Chen & Guestrin, 2016) on the generated synthetic datasets. Next, we conduct experiments on *low-order statistics* where we perform column-wise density estimation (CDE) and pair-wise column correlation (PCC). Lastly, we examine the models' quality on *high-order metrics* such as $\alpha$-*precision* and $\beta$-*recall* scores (Alaa et al., 2022). We add two extra benchmarks including a detection test metric, Classifier Two Sample Tests (C2ST) (SDMetrics, 2024) – Appendix D.6, and a privacy preservation metric, Distance to Closest Record (DCR) (Minieri, 2022) – Appendix D.7. Further details are found in Appendix C.4.

### 4.1 Model Comparisons on Predefined Baselines

We benchmark TabUnite(i2b)-Flow, TabUnite(dic)-Flow, and TabUnite(psk)-Flow across 6 datasets, against a wide range of baselines, in terms of a downstream MLE task. Following the setting in TabDDPM and TabSYN (Kotelnikov et al., 2023; Zhang et al., 2023), we split the datasets into training and testing sets where the generative models are trained on the training set. Synthetic samples of equivalent size are then generated based on the trained generative models. The generated data is subsequently evaluated against the mentioned benchmarks, using the testing set—unseen during training and generation phases—to assess the models' performance and generalization.

As observed in Table 2, TabUnite methods achieve the best MLE performance compared to existing baselines across 6 datasets. We also identify that PSK is the most superior encoding setup, obtaining the best results in 5/6 of the datasets. Furthermore, we observe that TabUnite(psk)-Flow yields the highest performance in 3/6 of the best-performing TabUnite models.

Table 2: AUC (classification) and RMSE (regression) scores of Machine Learning Efficiency. ↑ indicates that the higher the score, the better the performance, vice versa. Values bolded in **red** is the best-performing model. Details are found in Appendix C.

| Methods | Adult | Default | Shoppers | Magic | Beijing | News |
|---|---|---|---|---|---|---|
| | AUC ↑ | AUC ↑ | AUC ↑ | AUC ↑ | RMSE ↓ | RMSE ↓ |
| Real | $0.927_{\pm 0.000}$ | $0.770_{\pm 0.005}$ | $0.926_{\pm 0.001}$ | $0.946_{\pm 0.001}$ | $0.423_{\pm 0.003}$ | $0.842_{\pm 0.002}$ |
| SMOTE | $0.899_{\pm 0.007}$ | $0.741_{\pm 0.009}$ | $0.911_{\pm 0.012}$ | $0.934_{\pm 0.008}$ | $0.593_{\pm 0.011}$ | $0.897_{\pm 0.036}$ |
| CTGAN | $0.886_{\pm 0.002}$ | $0.696_{\pm 0.005}$ | $0.875_{\pm 0.009}$ | $0.855_{\pm 0.006}$ | $0.902_{\pm 0.019}$ | $0.880_{\pm 0.016}$ |
| TVAE | $0.878_{\pm 0.004}$ | $0.724_{\pm 0.005}$ | $0.871_{\pm 0.006}$ | $0.887_{\pm 0.003}$ | $0.770_{\pm 0.011}$ | $1.01_{\pm 0.016}$ |
| GOGGLE | $0.778_{\pm 0.012}$ | $0.584_{\pm 0.005}$ | $0.658_{\pm 0.052}$ | $0.654_{\pm 0.024}$ | $1.09_{\pm 0.025}$ | $0.877_{\pm 0.002}$ |
| GReaT | $0.844_{\pm 0.005}$ | $0.755_{\pm 0.006}$ | $0.902_{\pm 0.005}$ | $0.888_{\pm 0.008}$ | $0.653_{\pm 0.013}$ | OOM |
| STaSy | $0.906_{\pm 0.001}$ | $0.752_{\pm 0.006}$ | $0.914_{\pm 0.005}$ | $0.934_{\pm 0.003}$ | $0.656_{\pm 0.014}$ | $0.871_{\pm 0.002}$ |
| CoDi | $0.871_{\pm 0.006}$ | $0.525_{\pm 0.006}$ | $0.865_{\pm 0.006}$ | $0.932_{\pm 0.003}$ | $0.818_{\pm 0.021}$ | $1.21_{\pm 0.005}$ |
| TabDDPM | $0.910_{\pm 0.001}$ | $0.761_{\pm 0.004}$ | $0.915_{\pm 0.004}$ | $0.932_{\pm 0.003}$ | $0.592_{\pm 0.012}$ | $3.46_{\pm 1.25}$ |
| TabSYN[1] | $0.906_{\pm 0.001}$ | $0.755_{\pm 0.004}$ | $0.918_{\pm 0.004}$ | $0.935_{\pm 0.003}$ | $0.586_{\pm 0.013}$ | $0.862_{\pm 0.021}$ |
| ForestFlow[2] | OOM | OOM | $0.918_{\pm 0.003}$ | $0.936_{\pm 0.003}$ | OOM | OOM |
| TabUnite(i2b)-DDPM | $0.912_{\pm 0.001}$ | $0.762_{\pm 0.003}$ | $0.919_{\pm 0.004}$ | $\mathbf{0.944}_{\pm 0.002}$ | $0.542_{\pm 0.008}$ | $0.844_{\pm 0.013}$ |
| TabUnite(dic)-DDPM | $0.912_{\pm 0.002}$ | $0.763_{\pm 0.005}$ | $0.910_{\pm 0.006}$ | $0.943_{\pm 0.003}$ | $0.541_{\pm 0.005}$ | $0.851_{\pm 0.012}$ |
| TabUnite(psk)-DDPM | $\mathbf{0.913}_{\pm 0.002}$ | $0.764_{\pm 0.005}$ | $0.912_{\pm 0.005}$ | $0.939_{\pm 0.003}$ | $\mathbf{0.508}_{\pm 0.006}$ | $0.836_{\pm 0.0015}$ |
| TabUnite(i2b)-Flow | $0.911_{\pm 0.001}$ | $0.763_{\pm 0.004}$ | $0.918_{\pm 0.005}$ | $0.941_{\pm 0.003}$ | $0.543_{\pm 0.007}$ | $0.847_{\pm 0.014}$ |
| TabUnite(dic)-Flow | $0.911_{\pm 0.002}$ | $0.758_{\pm 0.006}$ | $0.908_{\pm 0.006}$ | $0.943_{\pm 0.003}$ | $0.555_{\pm 0.006}$ | $0.848_{\pm 0.013}$ |
| TabUnite(psk)-Flow | $0.912_{\pm 0.002}$ | $\mathbf{0.782}_{\pm 0.005}$ | $\mathbf{0.919}_{\pm 0.005}$ | $0.941_{\pm 0.003}$ | $0.536_{\pm 0.006}$ | $\mathbf{0.814}_{\pm 0.0015}$ |

[1] Despite numerous rerun attempts per TabSYN's repo, we cannot reproduce Adult ($.915_{\pm.002}$) and Shoppers ($.920_{\pm.005}$) that are $\geq$ than our results in TabSYN's paper.

[2] ForestFlow's training procedure is CPU-based, its architecture cannot handle datasets $> 21,000$ samples.

## 4.2 ABLATION STUDY: ENCODING SCHEME AND MODEL CHOICE

To further validate the effectiveness of TabUnite's encoding schemes, we conduct an ablation study to isolate the generative model while varying the encoding methods among PSK, Dictionary, Analog Bits, separate modeling, and one-hot encoding. We also perform the reverse, isolating the encoding schemes while varying the generative models between Flow Matching and DDPM. The real-world datasets we select for comparison in our main text are "Adult" and "News" since they have a good amount of samples, as well as a balanced set of continuous and categorical features. Results for the remaining datasets can be found in Appendix D.2.

**Curation of a Large-Scaled Mixed Synthetic Dataset.** While our experiments using publicly available datasets from the UCI machine learning repository (Dua & Graff, 2017), as well as other databases (Vanschoren et al., 2013) are well established, an issue is that they lack datasets with a large number of samples ($> 100$k) and mixed features ($> 15$ continuous and categorical features). Therefore, the need for curating publicly available large datasets with mixed features remains crucial for determining the effectiveness of our categorical encoding schemes. A considerably larger dataset is the US Census Data (1990) (Meek et al., 2001) which contains $2,458,285$ samples and $61$ features. However, these samples consist of only categorical variables. To incorporate continuous features, we begin by converting ordinal categorical features into continuous features. With the remaining non-ordinal categorical features, we select a subset and convert them to continuous using Frequency Encoding. Lastly, we leverage a synthetic data generation model (Chen et al., 2018; Si et al., 2024) to create continuous composite indicators (OECD et al., 2008) that can help capture interactions between different aspects of the data. The synthetic continuous data are then generated per the following two polynomials: $\text{Syn1} = \exp(x_i x_j)$ and $\text{Syn2} = \exp(\sum_{i=1}^{3}(x_i^2 - 4))$ before applying a logistic function $\frac{1}{1+logit(\mathbf{X})}$. Finally, we concatenate our synthesized continuous features with the categorical. We have now constructed a Census Synthetic dataset comprised of $41$ continuous features, $40$ categorical features, and $2,458,285$ samples. For a regression task, the label is "dIncome1" which is the annual income of an individual. Further details can be found in Appendix C.1.

**Ablation Analysis.** In Table 3, TabUnite encoding methods achieve the best overall performance across the datasets and benchmarks. Solely comparing the performance of our encoding methods, we observe that TabUnite(psk) does the best, followed by TabUnite(i2b), TabUnite(dic), separated (TabDDPM/TabFlow), then one-hot. Evaluating Flow Matching vs. DDPM, we observe that while

Table 3: AUC (classification), RMSE (regression), Column-Wise Density Estimation (CDE), Pair-Wise Column Correlation (PCC), $\alpha$-Precision, and $\beta$-Recall scores for our Census Synthetic, Beijing, and Adult datasets. $\uparrow$ indicates that the higher the score, the better the performance, vice versa. Values bolded in **red** is the best-performing model. Details are found in Appendix C.

| Methods | Census Synthetic | | | | |
| --- | --- | --- | --- | --- | --- |
| | RMSE $\downarrow$ | CDE $\uparrow$ | PCC $\uparrow$ | $\alpha \uparrow$ | $\beta \uparrow$ |
| TabDDPM | $0.204_{\pm 0.012}$ | $83.60_{\pm 0.01}$ | $86.06_{\pm 0.11}$ | $72.78_{\pm 0.10}$ | $0.09_{\pm 0.05}$ |
| oheDDPM | $0.954_{\pm 0.024}$ | $54.95_{\pm 0.02}$ | $50.43_{\pm 0.01}$ | $0.00_{\pm 0.00}$ | $0.00_{\pm 0.00}$ |
| TabUnite(i2b)-DDPM | $0.188_{\pm 0.004}$ | $86.39_{\pm 0.01}$ | $90.61_{\pm 0.58}$ | $85.76_{\pm 0.10}$ | $36.34_{\pm 0.01}$ |
| TabUnite(dic)-DDPM | $0.156_{\pm 0.005}$ | $\textbf{86.57}_{\pm \textbf{0.02}}$ | $90.85_{\pm 0.11}$ | $91.71_{\pm 0.02}$ | $36.27_{\pm 0.08}$ |
| TabUnite(psk)-DDPM | $0.171_{\pm 0.005}$ | $86.16_{\pm 0.01}$ | $90.51_{\pm 0.13}$ | $81.85_{\pm 0.08}$ | $37.37_{\pm 0.07}$ |
| TabFlow | $0.144_{\pm 0.005}$ | $85.80_{\pm 0.01}$ | $90.74_{\pm 0.70}$ | $94.12_{\pm 0.04}$ | $\textbf{42.06}_{\pm \textbf{0.10}}$ |
| oheFlow | $0.502_{\pm 0.003}$ | $64.29_{\pm 0.01}$ | $69.82_{\pm 0.19}$ | $67.09_{\pm 0.08}$ | $0.00_{\pm 0.00}$ |
| TabUnite(i2b)-Flow | $\textbf{0.127}_{\pm \textbf{0.003}}$ | $86.24_{\pm 0.02}$ | $\textbf{91.59}_{\pm \textbf{0.11}}$ | $90.51_{\pm 0.07}$ | $41.38_{\pm 0.07}$ |
| TabUnite(dic)-Flow | $0.159_{\pm 0.003}$ | $86.12_{\pm 0.02}$ | $91.53_{\pm 0.10}$ | $\textbf{95.64}_{\pm \textbf{0.06}}$ | $39.02_{\pm 0.05}$ |
| TabUnite(psk)-Flow | $0.156_{\pm 0.004}$ | $85.78_{\pm 0.02}$ | $90.90_{\pm 0.12}$ | $89.58_{\pm 0.08}$ | $40.17_{\pm 0.07}$ |

| Methods | Adult | | | | |
| --- | --- | --- | --- | --- | --- |
| | AUC $\uparrow$ | CDE $\uparrow$ | PCC $\uparrow$ | $\alpha \uparrow$ | $\beta \uparrow$ |
| TabDDPM | $0.909_{\pm 0.002}$ | $98.37_{\pm 0.05}$ | $96.69_{\pm 0.50}$ | $90.99_{\pm 0.37}$ | $\textbf{62.19}_{\pm \textbf{0.69}}$ |
| oheDDPM | $0.476_{\pm 0.057}$ | $48.54_{\pm 2.13}$ | $35.51_{\pm 2.67}$ | $11.01_{\pm 4.52}$ | $0.47_{\pm 0.07}$ |
| TabUnite(i2b)-DDPM | $0.912_{\pm 0.003}$ | $99.27_{\pm 0.13}$ | $\textbf{98.13}_{\pm \textbf{0.21}}$ | $98.50_{\pm 0.23}$ | $47.86_{\pm 0.09}$ |
| TabUnite(dic)-DDPM | $0.912_{\pm 0.002}$ | $98.97_{\pm 0.06}$ | $97.82_{\pm 0.19}$ | $98.36_{\pm 0.29}$ | $51.34_{\pm 0.20}$ |
| TabUnite(psk)-DDPM | $\textbf{0.913}_{\pm \textbf{0.002}}$ | $\textbf{99.24}_{\pm \textbf{0.06}}$ | $98.10_{\pm 0.37}$ | $97.99_{\pm 0.19}$ | $52.04_{\pm 0.34}$ |
| TabFlow | $0.908_{\pm 0.002}$ | $96.32_{\pm 0.52}$ | $93.76_{\pm 0.76}$ | $89.34_{\pm 3.61}$ | $52.71_{\pm 0.36}$ |
| oheFlow | $0.895_{\pm 0.003}$ | $91.05_{\pm 0.78}$ | $84.92_{\pm 0.87}$ | $93.55_{\pm 3.61}$ | $30.26_{\pm 0.76}$ |
| TabUnite(i2b)-Flow | $0.911_{\pm 0.001}$ | $98.47_{\pm 0.16}$ | $97.28_{\pm 0.27}$ | $99.10_{\pm 0.38}$ | $48.35_{\pm 0.34}$ |
| TabUnite(dic)-Flow | $0.910_{\pm 0.002}$ | $98.23_{\pm 0.22}$ | $96.01_{\pm 0.72}$ | $98.48_{\pm 0.68}$ | $51.10_{\pm 0.22}$ |
| TabUnite(psk)-Flow | $0.912_{\pm 0.001}$ | $98.75_{\pm 0.17}$ | $97.55_{\pm 0.40}$ | $\textbf{99.21}_{\pm \textbf{0.30}}$ | $49.68_{\pm 0.28}$ |

| Methods | News | | | | |
| --- | --- | --- | --- | --- | --- |
| | RMSE $\downarrow$ | CDE $\uparrow$ | PCC $\uparrow$ | $\alpha \uparrow$ | $\beta \uparrow$ |
| TabDDPM | $0.842_{\pm 0.025}$ | $94.79_{\pm 1.17}$ | $89.52_{\pm 2.74}$ | $90.94_{\pm 1.43}$ | $40.82_{\pm 0.52}$ |
| oheDDPM | $0.840_{\pm 0.020}$ | $98.06_{\pm 0.07}$ | $97.10_{\pm 0.44}$ | $96.31_{\pm 0.54}$ | $47.10_{\pm 0.29}$ |
| TabUnite(i2b)-DDPM | $0.844_{\pm 0.024}$ | $98.26_{\pm 0.04}$ | $\textbf{99.04}_{\pm \textbf{0.24}}$ | $96.09_{\pm 0.50}$ | $48.31_{\pm 0.40}$ |
| TabUnite(dic)-DDPM | $0.851_{\pm 0.019}$ | $98.22_{\pm 0.04}$ | $98.57_{\pm 0.27}$ | $96.95_{\pm 0.44}$ | $47.94_{\pm 0.34}$ |
| TabUnite(psk)-DDPM | $\textbf{0.836}_{\pm \textbf{0.021}}$ | $\textbf{98.35}_{\pm \textbf{0.04}}$ | $98.78_{\pm 0.18}$ | $95.11_{\pm 0.28}$ | $48.65_{\pm 0.33}$ |
| TabFlow | $0.850_{\pm 0.017}$ | $96.76_{\pm 0.18}$ | $97.98_{\pm 0.11}$ | $90.20_{\pm 1.04}$ | $50.13_{\pm 0.28}$ |
| oheFlow | $0.850_{\pm 0.022}$ | $96.09_{\pm 0.46}$ | $98.06_{\pm 0.13}$ | $\textbf{98.02}_{\pm \textbf{1.44}}$ | $43.38_{\pm 0.76}$ |
| TabUnite(i2b)-Flow | $0.848_{\pm 0.016}$ | $96.91_{\pm 0.09}$ | $98.43_{\pm 0.23}$ | $90.06_{\pm 0.48}$ | $\textbf{52.00}_{\pm \textbf{0.31}}$ |
| TabUnite(dic)-Flow | $0.853_{\pm 0.014}$ | $96.56_{\pm 0.35}$ | $98.13_{\pm 0.21}$ | $92.39_{\pm 1.28}$ | $50.52_{\pm 0.39}$ |
| TabUnite(psk)-Flow | $0.847_{\pm 0.014}$ | $96.89_{\pm 0.10}$ | $98.34_{\pm 0.36}$ | $90.91_{\pm 1.29}$ | $51.75_{\pm 0.54}$ |

[1] oheDDPM collapses on Census Synthetic and Adult for $\alpha$ and $\beta$.

DDPM outperforms FM in most scenarios, FM still remains competitive while providing gains in sampling speed and efficiency.

**Sampling Speed.** We investigate the sampling speed of Flow Matching against DDPM and DDIM. In Figure 3, we examine the number of function evaluations the methods require to converge to its best AUC and average error on the Adult dataset. We observe that TabUnite(psk)-Flow and TabFlow converge to their best AUC/Average Error in as little as 32 NFEs when compared to DDPM methods and TabDDIM that require around 1000 NFEs. Additionally, the performance of TabUnite(psk)-Flow remains competitive to TabUnite(psk)-DDPM at convergence. Therefore, Flow Matching is computationally efficient and fast at sampling while providing competitive results to DDPMs.

### 4.3 SUMMARY OF ADDITIONAL EXPERIMENTS

We present additional experiments in the Appendix to strengthen our findings.

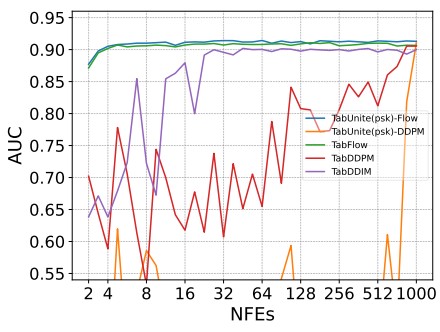 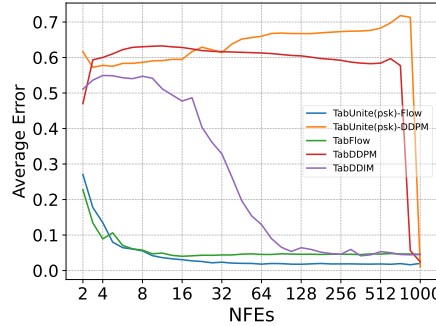

(a) AUC vs. Sampling Speed (NFEs)   (b) Avg. Error vs. Sampling Speed (NFEs)

Figure 3: Synthetic Data Quality vs. Sampling Speed of TabUnite(psk)-Flow/DDPM, TabFlow, Tab-DDPM, and TabDDIM on the Adult dataset. TabUnite(psk)-Flow converges to the best AUC/Average Error in much fewer NFEs compared to the baselines.

- Additional results for Tables 2 and 3 with more datasets can be found in Appendix D.1 and D.2. Their results further emphasize the importance and superiority of TabUnite.

- To justify the faithfulness of our model, we use synthetic toy examples to assess our model's integrity against the known ground truth. Details and results can be found in Appendix C.1 and D.3 respectively. The synthetic dataset illustrates that TabUnite methods are faithful in generating high-quality samples that match the ground truth qualitatively. Quantitatively, training TabUnite methods are stable and converge to a higher accuracy than TabDDPM.

- In Appendix D.4 and D.5, we present low-order statistics and high-order metrics results of our baselines. The consensus indicates that TabSYN is the best-performing model among the baselines. However, results in Table 3 and Appendix D.2 indicated that TabUnite models outperform TabSYN.

- In Appendix D.6 and Appendix D.7 we include results to a detection test metric, Classifier Two Sample Tests (C2ST) (SDMetrics, 2024), and a privacy preservation metric, Distance to Closest Record (DCR) (Minieri, 2022). C2ST results showcase that most TabUnite methods outperform the baselines, highlighting that the synthetic data is similar to the real data. However, we do not outperform TabSYN in our DCR results, indicating that the synthetic data will leak the real data's information. This aligns with our hypothesis where TabSYN leverages a latent space thus, resulting in a lossy compression, improving their DCR scores.

- In addition to generation, we also conducted a brief investigation on the effects of TabUnite encoding on prediction tasks in Appendix D.8. While the improvements are less significant than those of the generation, TabUnite encoding schemes still outperform one-hot encoding.

## 5 CONCLUSION

We propose an efficient encoding framework for tabular flow and diffusion models that leverages effective categorical encoding schemes to unify the data space. Since flow and diffusion models rely on continuous transformations of denoising score-matching, or invertible mappings between the data and latent space, applying a single flow/diffusion model that captures heterogeneous feature interrelationships is crucial in improving generation quality. Our models are curated by employing PSK, Dictionary, and Analog Bits encoding that efficiently convert categorical variables into a dense and meaningful continuous representation, before applying Conditional Flow Matching to generate the data. To further strengthen our findings on our categorical embedding schemes, we curate a large-scale heterogeneous tabular dataset and benchmark TabUnite on it. Relative to the baselines, our TabUnite models, notably TabUnite(psk)-Flow, outperform them across a wide range of datasets while evaluated on a broad suite of benchmarks. Additionally, leveraging Flow Matching greatly bolsters our sampling efficiency, saving computational cost and time while remaining competitive in performance to DDPMs. Overall, we justify our claim of applying efficient encoding methods such as TabUnite(psk) to a single efficient flow matching model on a coherent data space.

# Appendix

CONTENTS

## A  ALGORITHMS

Algorithms 1 and 2 describe the training and sampling of TabUnite's Flow Matching process. For more information regarding Flow Matching, please refer to "Flow Matching for Generative Modeling" (Lipman et al., 2022) or "Improving and Generalizing Flow-Based Generative Models with Minibatch Optimal Transport" (Tong et al., 2023).

---

**Algorithm 1** TabUnite: Training Flow Matching using CFM

---

1: Sample initial data points $x_1 \sim q(x_1)$
2: Initialize vector field $v_t(x)$ and parameters $\theta$
3: **while** not converged **do**
4:      Sample time step $t \sim U([0, 1])$
5:      Sample $x \sim p_t(x|x_1)$
6:      Calculate true vector field $u_t(x|x_1)$ as per Eq. 11
7:      Compute loss $L_{CFM}(\theta) = \mathbb{E}|v_t(x) - u_t(x|x_1)|^2$
8:      Update $\theta$ using gradient descent to minimize $L_{CFM}(\theta)$
9: **end while**

---

---

**Algorithm 2** TabUnite: Sampling Flow Matching using CFM

---

1: Sample $x \sim \mathcal{N}(\mathbf{0}, \mathbf{I})$ (start with the noise distribution)
2: Set $t_{\max} = T$ and initialize $x_T = x$
3: **for** $i = T, \dots, 1$ **do**
4:      Use $\psi_t$ to map $x_T$ to $x_{t_{i-1}}$ using the learned vector field $u_t$
5:      Compute $x_{t_{i-1}}$ with $\psi_{t_i}(x_T) = \sigma_{t_i}(x_1)x_T + \mu_{t_i}(x_1)$
6:      Update $x_T = x_{t_{i-1}}$
7: **end for**
8: $x_0$ is a synthetic sample generated by CFM

---

# B ARCHITECTURE

## B.1 FLOW MATCHING/DDPM MLP

Figure 4 illustrates the MLP architecture used as part of our Flow Matching network, also used in TabDDPM (Kotelnikov et al., 2023) and TabSYN (Zhang et al., 2023), which is based on (Gorishniy et al., 2023).

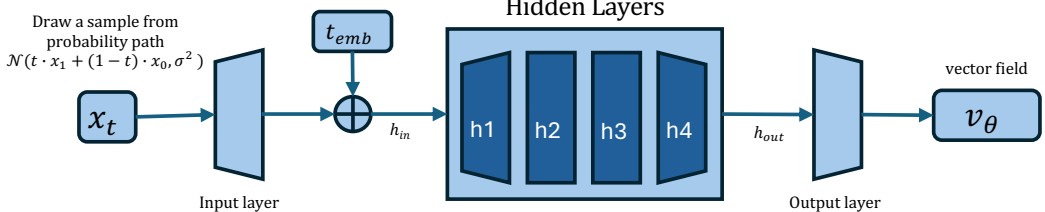

Figure 4: The MLP architecture used in the Flow Matching.DDPM process. The neural network takes in a batch of samples drawn from the probability path at time $t$'s sampled from $\mathcal{U}(0,1)$ to create a vector field $v_\theta$ that represents a continuous normalizing flow from pure noise to our data distribution $p_1(x_1)$.

The input layer projects the batch of tabular data input samples $x_t$, each with dimension $d_{in}$, to the dimensionality $d_t$ of our time step embeddings $t_{emb}$ through a fully connected layer. This is so that we may leverage temporal information, which is appended to the result of the projection in the form of sinusoidal time step embeddings.

$$h_{in} = FC_{d_t}(x_t) + t_{emb} \qquad (6)$$

The hidden layers $h1$, $h2$, $h3$, and $h4$ are fully connected networks used to learn and create the vector field. The output dimension of each layer is chosen as $d_t$, $2d_t$, $2d_t$, and $d_t$ respectively. On top of the FC networks, each layer also consists of an activation function followed by dropout, as seen in the formulas below. This formulation is repeated for each hidden layer, at the end of which we obtain $h_{out}$. The exact activations, dropout, and other hyperparameters chosen are shown in Table 4.

$$h_1 = \text{Dropout}(\text{Activation}(FC(h_{in}))) \qquad (7)$$

At last, the output layer transforms $h_{out}$, of dimension $t_{emb}$ back to dimension $d_{in}$ through a fully connected network, which now represents the vector field $v_\theta$.

$$v_\theta = FC_{d_{in}}(h_{out}) \qquad (8)$$

## B.2 FURTHER FLOW MATCHING DETAILS

Let $x$ denote a sample from the dataset $\mathbf{X}_{unite}$, i.e. $x \sim \mathbf{X}_{unite}$. We learn a vector field $v_t(x)$ to approximate the true vector field $u_t(x|x_1)$, yielding an objective function of the following:

$$L_{CFM}(\theta) = \mathbb{E}_{q(x_1),p_t(x|x_1)}||v_t(x) - u_t(x|x_1)||^2 \qquad (9)$$

This in turn, generates a probability density path $p_t(x|x_1)$ where the density evolves from the initial standard normal distribution $p_0(x|x_1) = p_0(x) = N(x|0,I)$, and ultimately converges to the underlying data distribution $p_1(x|x_1)$ centered around $x = x_1$ where we sample $x_{syn}$. In order to generate the aforementioned path $p_t(x|x_1)$ via vector field $u_t(x|x_1)$, we consider the flow $\psi_t$:

$$[\psi_t]_*p(x) = p_t(x|x_1) \qquad (10)$$

where $\psi_t(x) = \sigma(x_1)x + \mu_t(x_1)$. This property helps move the noise distribution from $p_0(x|x_1) = p(x)$ to $p_t(x|x_1)$. In other words, $\psi_t$ provides a vector field:

$$\frac{d}{dt}\psi_t(x) = u_t(\psi_t(x)|x_1) \qquad (11)$$

that generates the conditional probability path. Rewriting the objective function and reparameterizing in terms of $x_0$, we have:

$$L_{CFM}(\theta) = \mathbb{E}_{q(x_1), p_t(x|x_1)} ||v_t(\psi(x)) - u_t(\psi_t(x)|x_1)||^2 \tag{12}$$

$$L_{CFM}(\theta) = \mathbb{E}_{q(x_1), p_t(x|x_1)} ||v_t(\psi(x_0)) - \frac{d}{dt}\psi_t(x_0)||^2 \tag{13}$$

With the simple affine map property of $\psi_t$, we use it to solve for vector field $u$:

$$u_t(x|x_1) = \frac{\sigma_t'(x_1)}{\sigma_t(x_1)}(x - \mu_t(x_1)) + \mu_t'(x_1) \tag{14}$$

generating Gaussian probability path $p_t(x|x_1)$. Lastly, by integrating optimal transport theories, the final objective function is the following:

$$L_{CFM}(\theta) = \mathbb{E}_{t, q(x_1), p(x_0)} ||v_t(\psi_t(x_0)) - (x_1 - (1 - \sigma_{min})x_0)||^2 \tag{15}$$

## B.3 HYPERPARAMETERS

We generally utilise the same hyperparameters as TabSYN (Zhang et al., 2023) and TabDDPM (Kotelnikov et al., 2023) for comparability. The exact hyperparameters selected for our models are shown below in Table 4.

Table 4: TabUnite Hyperparameters.

| General | | Flow Matching/DDPM MLP | |
|---|---|---|---|
| **Hyperparameter** | **Value** | **Hyperparameter** | **Value** |
| Training Iterations | $100,000$ | Timestep embedding dimension $d_t$ | 1024 |
| Flow Matching/DDPM Sampling Steps | $50/1000$ | Activation | ReLU |
| Learning Rate | $1e-4$ | Dropout | 0.0 |
| Weight Decay | $5e-4$ | Hidden layer dimension $[h1, h2, h3, h4]$ | $[1024, 2048, 2048, 1024]$ |
| Batch Size | 4096 | | |

## C  EXPERIMENTAL DETAILS

The following delineates the foundation of our experiments:

- Codebase: Python & PyTorch
- CPU: AMD Threadripper 3960X
- GPU: Nvidia RTX A6000, 24GB VRAM
- Optimizer: Adam (Kingma & Ba, 2014)

### EXPERIMENT TABLE DETAILS

In Table 2, Table 3, and Appendix Tables, all reported results of baselines in our experiments are taken from (Zhang et al., 2023), except for TabSYN and TabDDPM, whose results are reproduced utilizing the public repository: `https://github.com/amazon-science/tabsyn`, following their recommended hyperparameters. For Flow Matching experiments, we maintained the same training steps of 100k while sampling for 50 steps. Additionally, for Table 2, we decided to rerun GReaT in the same original setting (1 Train, 20 Samples) for the Adult dataset as TabSYN's reported results $(0.913 \pm 0.003)$ were unusually high. All reported results follow TabSYN's 1 Training and 20 Sampling trial setting. Note that TabDDPM collapses on the News dataset for all the benchmarks.

For the "Census Synthetic" dataset, all metrics are evaluated on a 10% subsample. The reason is that it is computationally costly to compute results for diffusion-based models.

### C.1  DATASETS

### REAL WORLD DATASETS

Experiments were conducted with a total of 6 tabular datasets from the UCI Machine Learning Repository (Dua & Graff, 2017) with a (CC-BY 4.0) license. Classification tasks were performed on the Adult, Default, Magic, and Shoppers datasets, while regression tasks were performed on the Beijing and News datasets. Each dataset was split into training, validation, and testing sets with a ratio of 8:1:1, except for the Adult dataset, whose official testing set was used and the remainder split into training and validation sets with an 8:1 ratio. The resulting statistics of each dataset are shown below in Table 5. Note that the target column indicates the specific operation applied to each dataset: binary classification for a categorical target with two classes, multiclass classification for a categorical target with more than two classes, and regression for a numerical target feature. Some detailed information as well as the statistics of the datasets are shown in Tables 5 and 6 respectively.

Table 5: Statistics of datasets. "# Num" stands for the number of numerical columns, and "# Cat" stands for the number of categorical columns.

| Dataset | # Rows | # Num | # Cat | # Train | # Validation | # Test | Task Type |
|---|---|---|---|---|---|---|---|
| **Adult** | $48,842$ | $6$ | $9$ | $28,943$ | $3,618$ | $16,281$ | Binary Classification |
| **Default** | $30,000$ | $14$ | $11$ | $24,000$ | $3,000$ | $3,000$ | Binary Classification |
| **Shoppers** | $12,330$ | $10$ | $8$ | $9,864$ | $1,233$ | $1,233$ | Binary Classification |
| **Magic** | $19,019$ | $10$ | $1$ | $15,215$ | $1,902$ | $1,902$ | Binary Classification |
| **Beijing** | $41,757$ | $7$ | $5$ | $33,405$ | $4,175$ | $4,175$ | Regression |
| **News** | $39,644$ | $46$ | $2$ | $31,714$ | $3,965$ | $3,965$ | Regression |
| **Bank** | $45,211$ | $7$ | $10$ | $36,168$ | $4,521$ | $4,521$ | Binary Classification |
| **Cardio** | $70,000$ | $5$ | $7$ | $56,000$ | $7,000$ | $7,000$ | Binary Classification |
| **Stroke** | $4,909$ | $3$ | $8$ | $3,927$ | $490$ | $490$ | Binary Classification |
| **Census Synthetic** | $2,458,285$ | $41$ | $40$ | $1,966,621$ | $245,827$ | $245,829$ | Regression |

### SYNTHETIC TOY DATASETS

*Qualitative Toy Dataset:* The dataset consists of four columns, with the first two columns representing numerical data point coordinates. Subsequently, the third column categorizes the data points into five circles whereas the last column indicates the 5 colours each data point can be classified into.

Table 6: Details of datasets. The "Feature Information" column details the contents of the dataset and how it is curated. The "Prediction Task" column describes the model's objective on that dataset.

| Dataset | Feature Information | Prediction Task |
|---|---|---|
| **Adult** | Demographic and occupational variables from census data | Whether an individual's income exceeds $50,000 |
| **Default** | Demographic and account-specific data collected from credit card clients | Whether an individual will default on their credit card payments next month |
| **Shoppers** | Internet users' browser session information | Whether the user will engage in online shopping |
| **Magic** | Generated events simulating the imaging of gamma-ray air showers | Predict the type of high-energy gamma particles in the atmosphere |
| **Beijing** | Hourly atmospheric PM2.5 and meteorological data readings at the U.S. Embassy in Beijing | Predict future PM2.5 readings |
| **News** | Various features from the news site Mashable's published articles | The number of "shares" articles will have on social media |
| **Census Synthetic** | 1990 Census Demographics of the US Population | Annual Income of an individual |

Therefore, each row in the dataset contains 2 numerical features and 2 categorical features. A total of $10,000$ samples are generated for this dataset.

*Quantitative Toy Dataset:* To quantify our model's ability to generate high-quality data, we generate a synthetic toy dataset with 11 numerical features, all drawn from a unit Gaussian distribution, to represent a complex underlying data distribution. From these numerical features, we derive six categorical variables by applying a variety of transformations, the details of which are described by the equations below.

$$
\begin{aligned}
x_1^{cat} &= x_0^{num} \cdot x_1^{num} \\
x_2^{cat} &= (x_2^{num})^2 + (x_3^{num})^2 + (x_4^{num})^2 + (x_5^{num})^2 - 4 \\
x_3^{cat} &= -10 \cdot \sin(2 \cdot x_6^{num}) + 2 \cdot |x_7^{num}| + x_8^{num} - e^{-x_9^{num}} \\
x_4^{cat} &= (x_9^{num} < 0) \cdot x_1^{cat} + (1 - (x_9^{num} < 0)) \cdot x_2^{cat} \\
x_5^{cat} &= (x_9^{num} < 0) \cdot x_1^{cat} + (1 - (x_9^{num} < 0)) \cdot x_3^{cat} \\
x_6^{cat} &= (x_9^{num} < 0) \cdot x_2^{cat} + (1 - (x_9^{num} < 0)) \cdot x_3^{cat}
\end{aligned}
\tag{16}
$$

Following the transformations, tanh activation functions are applied followed by digitization to 10 separate bins. A total of $10,000$ samples are generated for this dataset, resulting in our discrete categorical variables. We quantify the performance of our models by examining the fidelity of generating these categorical variables. The scoring is determined by taking the absolute value of the difference between the real and synthesized values.

We perform three trial experiments for each method and report their mean and standard deviation. Note that in the quantitative experiments, we use a DDIM sampler for TabDDPM thus, the results are slightly worse than those we reported in our previous tables.

CENSUS SYNTHETIC DATASET

The US Census Data (1990) (Meek et al., 2001) ((CC-BY 4.0) license) contains $2,458,285$ samples and 61 features (excluding "dIncome2" to "dIncome8" since they are redundant). However, these samples consist of only categorical variables. To incorporate continuous features, we begin by converting the following ordinal categorical features into continuous features:

- Annual income: dIncome1
- Earnings from employment: dRearning
- Age: dAge

- English proficiency: iEnglish
- Hours worked in 1989: dHour89
- Hours worked per week: dHours
- Travel time to work: dTravtime
- Years spent schooling: iYearsch
- Years spent working: iYearwrk

A total of 9 ordinal categorical features are converted. With the remaining non-ordinal categorical features, we select 12 additional categorical features and convert them to continuous using Frequency Encoding yielding us 21 continuous features in total. We consider features that are likely to have a variety of categories and could benefit from a frequency-based transformation. For instance, occupation covers a wide range of jobs and ancestry covers many different backgrounds. The features are as follows:

- Primary ancestry: dAncstry1
- Secondary ancestry: dAncstry2
- Citizenship status: iCitizen
- Marital status: iMarital
- Hispanic origin: dHispanic
- Class of worker: iClass
- Place of birth: dPOB
- Occupation: dOccup
- Industry: dIndustry
- Mobility status: iMobility
- Relationship to head of household: iRelat1
- Sex: iSex

Lastly, to balance out the remaining categorical features 40 with the 21 continuous ones, we leverage a synthetic data generation model (Chen et al., 2018; Yoon et al., 2019; Si et al., 2024) to generate 20 more continuous features based on the converted continuous features. We create continuous composite indicators (OECD et al., 2008) by combining our curated continuous features in sets of 2 or 3 that can help capture interactions and relationships between different aspects of the data. An example is a gender and earnings indicator that shows income disparities. Here are the composite indicators:

- Work hours (Hours worked per week and Hours worked in 1989): dHours, dHour89
- Educational attainment with age (Age and Years of schooling): dAge, iYearsch
- Language skills based on birthplace (English proficiency and Place of birth): iEnglish, dPOB
- Demographic relationships (Citizenship status and Hispanic origin): iCitizen, dHispanic
- Commuting patterns (Travel time to work and Years worked): dTravtime, iYearwrk
- Family structure (Marital status and Relationship to household head): iMarital, iRelat1
- Employment characteristics (Industry and Occupation): dIndustry, dOccup
- Income disparities (Gender and Earnings): iSex, dRearning
- Migration patterns (Mobility status and Citizenship): iMobility, iCitizen
- Heritage (Primary and Secondary Ancestry): dAncstry1, dAncstry2
- Career dedication (Hours worked per week, Hours worked in 1989, and Travel time to work): dHours, dHour89, dTravtime
- Career progression (Age, years of schooling, and years worked): dAge, iYearsch, iYearwrk
- Cultural integration (English proficiency, place of birth, and citizenship): iEnglish, dPOB, iCitizen

- Household dynamics (Marital status, relationship to household head, and mobility status): iMarital, iRelat1, iMobility

- Job characteristics (Industry, Occupation, and Earnings): dIndustry, dOccup, dRearning

- Income trends (Gender, Earnings, and Age): iSex, dRearning, dAge

- Heritage and immigration status (Primary and Secondary heritage, and Citizenship): dAncstry1, dAncstry2, iCitizen

- Demographic patterns (Hispanic origin, Relationship to household head, and Age): dHispanic, iRelat1, dAge

- Job location and stability (Travel time, Years worked, and Occupation): dTravtime, iYearwrk, dOccup

- Education's impact on earnings (Years of schooling, Years worked, and Earnings): iYearsch, iYearwrk, dRearning

Before generating these composite indicators, we first apply a Standard scaler to the converted continuous features since the input features are "generated from a Gaussian distribution ($X \sim N(0, I)$)" (per (Chen et al., 2018)). The synthetic continuous data are then generated according to the following two polynomials:

- $\text{Syn1} = \exp(\mathbf{X}_i \mathbf{X}_j)$
- $\text{Syn2} = \exp(\sum_{i=1}^{3}(\mathbf{X}_i^2 - 4))$

where the first set consists of 10 indicators derived from pairs of variables following Syn1 and the second set consists of 10 indicators derived from triples of variables following Syn2. These composite indicators are then transformed using the logistic function $\frac{1}{1+\exp(\mathbf{X})}$. Finally, we merge our continuous features with the categorical features to create a comprehensive "Census Synthetic" dataset. The "Census Synthetic" dataset we construct comprises of $41$ continuous features, $40$ categorical features and $2,458,285$ samples. For a regression task, the label is "dIncome1" which is the Annual income of an individual. Note that the dataset will be released with a CC-BY 4.0 license.

## C.2 ADDITIONAL DETAILS ON BASELINES: PREDEFINED MODELS.

TabUnite's performance is evaluated in comparison to previous works in mixed-type tabular data generation. This includes CTGAN and TVAE (Xu et al., 2019), GOGGLE (Liu et al., 2023), GReaT (Borisov et al., 2023), STaSy (Kim et al., 2022), CoDi (Lee et al., 2023), TabDDPM (Kotelnikov et al., 2023), and TabSYN (Zhang et al., 2023). The underlying architectures and implementation details of these models are presented below in Table 8.

Table 8: Comparison of previous methods in Tabular Data Synthesis.

| Method | Model[1] | Type[2] | Categorical Encoding | Numerical Encoding | Additional Techniques |
|--------|----------|---------|---------------------|--------------------|-----------------------|
| **CTGAN** | GAN | U | One-Hot Encoding | Scaled Bayesian Gaussian Mixture | Mode-specific normalization to represent complex distributions & conditional generation to address data imbalances |
| **TVAE** | VAE | U | One-Hot Encoding | Scaled Bayesian Gaussian Mixture | Mode-specific normalization & conditional generation |
| **GOGGLE** | VAE + GNN | U | One-Hot Encoding | - | Learning relational structures among features graphically through an adjacency matrix |
| **GReaT** | Autoregressive GPT | U | Byte-Pair Encoding[3] | Byte-Pair Encoding[3] | Textual Encoder which converts data into natural language, followed by Feature Order Permutation and Fine-tuning |
| **STaSy** | Score-based Diffusion | U | One-Hot Encoding | Min-max scaler | Self-paced learning and fine-tuning |
| **CoDi** | DDPM/ Multinomial Diffusion | S | One-Hot Encoding | Min-max scaler | Model Inter-conditioning and Contrastive learning to learn dependencies between categorical and numerical data |
| **TabDDPM** | DDPM/ Multinomial Diffusion | S | One-Hot Encoding | Quantile Transformer | Concatenation of numerical and categorical features |
| **TabSYN** | VAE + EDM | U | One-Hot | Quantile Transformer | Feature Tokenizer and Transformer encoder to learn cross-feature relationships with adaptive loss weighing to increase reconstruction performance |
| **TabUnite-i2BFlow** | Flow Matching | U | Analog Bits | Quantile Transformer | Concatenation of numerical and categorical features encoded with TabUnite's embedding scheme |
| **TabUnite-dicFlow** | Flow Matching | U | Dictionary | Quantile Transformer | Concatenation of numerical and categorical features encoded with TabUnite's embedding scheme |

[1] The 'Model' Column indicates the underlying architecture used for the model. Options include Generative Adversarial Networks or GANs (Goodfellow et al., 2014), Variational Autoencoders or VAEs (Kingma & Welling, 2013), Denoising Diffusion Probabilistic Models or DDPMs (Ho et al., 2020b), Multinomial Diffusion (Hoogeboom et al., 2021), EDM, as introduced in (Karras et al., 2022).

[2] The 'Type' column indicates the data integration approach used in the model. 'U' denotes a unified data space where numerical and categorical data are combined after initial processing and fed collectively into the model. 'S' represents a separated data space, where numerical and categorical data are processed and fed into distinct models.

[3] Byte-Pair Encoding (Sennrich et al., 2016) is a tokenization method that iteratively merges the most frequent adjacent characters or character pairs into single tokens, creating a vocabulary of subwords that efficiently handles rare and unknown words in text processing.

### C.3 ADDITIONAL DETAILS ON ABLATIONS: ENCODING SCHEMES AND GENERATIVE MODELS (FLOW/DIFFUSION).

On top of the models developed by previous related works in mixed-type tabular data synthesis, we developed baselines that would provide a direct and analogous comparison to justify flow-matching and our particular encoding methods. This includes the flow-matching-based one-hotFlow (oheFlow), TabFlow, and the DDPM-based i2bDDPM, dicDDPM, and one-hotDDPM (oheDDPM).

Our DDPM-based baseline methods (i2bDDPM, dicDDPM, and oheDDPM) primarily inherit the design and implementation of TabDDPM (Kotelnikov et al., 2023). Whereas TabDDPM leverages two separate diffusion models, namely Gaussian diffusion and Multinomial diffusion, we devise i2bDDPM, dicDDPM, and oheDDPM to rely solely on Gaussian Diffusion. This is because their corresponding methods of Analog Bits, Dictionary Encoding, and One-Hot Encoding allow us to perform diffusion in a unified data space. Implementation of these methods is done by simply altering the data processing stage of the model. The DDPM architecture is largely kept the same.

Our Flow-based baseline methods (oheFlow, TabFlow) are extended from the TabUnite architecture, which consists of i2bFlow and dicFlow. oheFlow, as the name suggests, utilizes One-Hot Encoding in its data processing stage. Tabflow, on the other hand, mirrors the idea of TabDDPM in that two separate models are used: one for learning categorical features and the other for learning numerical features. Here, the implementation combines ordinary Flow Matching (Lipman et al., 2022) with Discrete Flow Matching (Campbell et al., 2024). The respective results of these two models are concatenated afterward to allow for the synthesis of mixed-type tabular data.

These methods all utilize the QuantileTransformer (Pedregosa et al., 2011) to process numerical data, which normalizes features to follow a uniform or normal distribution. This is done through sorting and ranking data points, and then mapping them to fit to the target distribution.

### C.4 BENCHMARKS

In this section, we expand on the concrete formulations behind our benchmarks including machine learning efficiency, low-order statistics, and high-order metrics. We also provide an overview on the detection and privacy metrics used in our experiments. These comprehensive benchmarks as well as their implementations are identical to those established by TabSYN (Zhang et al., 2023), ensuring a direct and accurate comparison.

#### MACHINE LEARNING EFFICIENCY

*AUC* (Area Under Curve) is used to evaluate the efficiency of our model in binary classification tasks. It measures the area under the Receiver Operating Characteristic (or ROC) curve, which plots the True Positive Rate against the False Positive Rate. AUC may take values in the range [0,1]. A higher AUC value suggests that our model achieves a better performance in binary classification tasks and vice versa.

$$\text{AUC} = \int_0^1 \text{TPR}(\text{FPR}) \, d(\text{FPR}) \tag{17}$$

*RMSE* (Root Mean Square Error) is used to evaluate the efficiency of our model in regression tasks. It measures the average magnitude of the deviations between predicted values ($\hat{y}_i$) and actual values ($y_i$). A smaller RMSE model indicates a better fit of the model to the data.

$$\text{RMSE} = \sqrt{\frac{1}{n} \sum_{i=1}^{n} (y_i - \hat{y}_i)^2} \tag{18}$$

#### LOW-ORDER STATISTICS.

*Column-wise Density Estimation* between numerical features is achieved with the Kolmogorov-Smirnov Test (KST). The Kolmogorov-Smirnov statistic is used to evaluate how much two underlying one-dimensional probability distributions differ, and is characterized by the below equation:

$$\text{KST} = \sup_x |F_1(x) - F_2(x)|, \tag{19}$$

where $F_n(x)$, the empirical distribution function of sample n is calculated by

$$F_n(x) = \frac{1}{n} \sum_{i=1}^{n} \mathbf{1}_{(-\infty, x]}(X_i) \tag{20}$$

*Column-wise Density Estimation* between two categorical features is determined by calculating the Total Variation Distance (TVD). This statistic captures the largest possible difference in the probability of any event under two different probability distributions. It is expressed as

$$TVD = \frac{1}{2} \sum_{x \in X} |P_1(x) - P_2(x)|, \tag{21}$$

where $P_1(x)$ and $P_2(x)$ are the probabilities (PMF) assigned to data point x by the two sample distributions respectively.

*Pair-wise Column Correlation* between two numerical features is computed using the Pearson Correlation Coefficient (PCC). It assigns a numerical value to represent the linear relationship between two columns, ranging from -1 (perfect negative linear correlation) to +1 (perfect positive linear correlation), with 0 indicating no linear correlation. It is computed as:

$$\rho(x, y) = \frac{\text{cov}(x, y)}{\sigma_x \sigma_y}, \tag{22}$$

To compare the Pearson Coefficients of our real and synthetic datasets, we quantify the dissimilarity in pair-wise column correlation between two samples

$$\text{Pearson Score} = \frac{1}{2} \mathbb{E}_{x,y} |\rho^1(x, y) - \rho^2(x, y)| \tag{23}$$

*Pair-wise Column Correlation* between two categorical features in a sample is characterized by a Contingency Table. This table is constructed by tabulating the frequencies at which specific combinations of the levels of two categorical variables work and recording them in a matrix format.

To Quantify the dissimilarity of contingency matrices between two different samples, we use the notion of the Contingency Score.

$$\text{Contingency Score} = \frac{1}{2} \sum_{\alpha \in A} \sum_{\beta \in B} |P_{1,(\alpha,\beta)} - P_{2,(\alpha,\beta)}|, \tag{24}$$

where $\alpha$ and $\beta$ describe possible categorical values that can be taken in features $A$ and $B$. $P_{1,(\alpha,\beta)}$ and $P_{2,(\alpha,\beta)}$ refer to the contingency tables representing the features $\alpha$ and $\beta$ in our two samples, which in this case corresponds to the real and synthetic datasets.

In order to obtain the column-wise density estimation and pair-wise correlation between a categorical and a numerical feature, we bin the numerical data into discrete categories before applying TVD and Contingency score respectively to obtain our low-order statistics.

We utilize the implementation of these experiments as provided by the SDMetrics library[1].

HIGH-ORDER STATISTICS

We utilize the implementations of High-Order Statistics as provided by the synthcity[2] library.

*α-precision* measures the overall fidelity of the generated data and is an extension of the classical machine learning quality metric of "precision". This formulation is based on the assumption that $\alpha$ fraction of our real samples are characteristic of the original data distribution and the rest are outliers. $\alpha$-precision therefore quantifies the percentage of generated synthetic samples that match $\alpha$ fraction of real samples (Alaa et al., 2022).

---

[1]https://github.com/sdv-dev/SDMetrics
[2]https://github.com/vanderschaarlab/synthcity

$\beta$-*recall* characterizes the diversity of our synthetic data and is similarly based on the quality metric of "recall". $\beta$-recall shares a similar assumption as $\alpha$-precision, except that we now assume that $\beta$ fraction of our synthetic samples are characteristic of the distribution. Therefore, this measure obtains the fraction of the original data distribution that is represented by the $\beta$ fraction of our generated samples (Alaa et al., 2022).

### DETECTION METRIC: CLASSIFIER TWO-SAMPLE TEST (C2ST)

The Classifier Two-Sample Test, a detection metric, assesses the ability to distinguish real data from synthetic data. This is done through a machine learning model that attempts to label whether a data point is synthetic or real. The score ranges from 0 to 1 where a score closer to 1 is superior, as it indicates that the machine learning model cannot concretely identify whether the data point in question is real or generated. We select logistic regression as our machine learning model in this case, using the implementation provided by SDMetric (SDMetrics, 2024).

### PRIVACY METRIC: DISTANCE TO CLOSEST RECORD (DCR)

The Distance to Closest Record metric quantifies the distance between each generated sample to our training set. The score is calculated as the proportion of synthetic data points that have a closer match to the real data set compared to the holdout set. A score close to 50% is ideal, as it indicates that our generated sample represents the underlying distribution of our training samples without revealing specific points present in the dataset.

## D  FURTHER EXPERIMENTAL RESULTS

We run all experiments outlined in this section on at least: TabUnite(i2b)-Flow/DDPM, TabUnite(dic)-Flow/DDPM, TabUnite-(psk)-Flow/DDPM, TabSYN (Zhang et al., 2023), and TabDDPM (Kotelnikov et al., 2023) due to their competitive performance in our MLE experiments, as seen in Table 2 as well as prior literature (Zhang et al., 2023). Unless otherwise stated, we use experimental results collected by TabSYN's author for all other model benchmarks. The metrics and error bars shown in the tables in this section are derived from the mean and standard deviation of the experiments performed on 20 randomly sampled sets of synthetic data.

### D.1  TABLE 1 RESULTS ON ADDITIONAL DATASETS

We include 3 more Kaggle datasets to obtain MLE results: Bank, Cardio, and Stroke. Bank contains 45,211 samples, 10 cat. features and 7 cont. features. Cardio contains 70,000 samples, 7 cat. features and 5 cont. features. Stroke contains 5,110 samples, 8 cat. features and 3 cont. features. The following are our results produced under the same setting as our Table 1. We include TabSYN and TabDDPM as baselines since they are the most competitive and relevant for us.

Table 10: AUC (classification) scores of Machine Learning Efficiency.

| Methods | Bank | Cardio | Stroke |
|---|---|---|---|
| | AUC ↑ | AUC ↑ | AUC ↑ |
| TabDDPM | **0.923±0.002** | 0.800±0.001 | 0.755±0.033 |
| TabSYN | 0.917±0.002 | 0.799±0.001 | 0.752±0.035 |
| TabUnite(i2b)-Flow | 0.919±0.002 | **0.801±0.001** | **0.763±0.031** |
| TabUnite(dic)-Flow | 0.920±0.002 | **0.801±0.001** | 0.743±0.037 |
| TabUnite(psk)-Flow | 0.918±0.001 | **0.801±0.001** | 0.762±0.027 |

We achieve the best results for Cardio and Stroke and the second best for Bank.

## D.2 TABLE 2 RESULTS ON ADDITIONAL DATASETS

| Methods | Beijing | | | | |
|---|---|---|---|---|---|
| | RMSE ↓ | CDE ↑ | PCC ↑ | $\alpha$ ↑ | $\beta$ ↑ |
| TabDDPM | $0.598_{\pm0.009}$ | $98.55_{\pm0.06}$ | $\mathbf{97.60}_{\pm0.2}$ | $98.35_{\pm0.27}$ | $55.66_{\pm0.24}$ |
| oheDDPM | $2.143_{\pm0.339}$ | $45.71_{\pm1.71}$ | $36.24_{\pm3.5}$ | $0.018_{\pm0.36}$ | $0.018_{\pm0.94}$ |
| TabUnite(i2b)-DDPM | $0.542_{\pm0.010}$ | $81.68_{\pm0.17}$ | $68.73_{\pm0.3}$ | $97.54_{\pm0.62}$ | $59.53_{\pm0.25}$ |
| TabUnite(dic)-DDPM | $0.541_{\pm0.007}$ | $\mathbf{98.96}_{\pm0.04}$ | $97.36_{\pm0.2}$ | $\mathbf{99.46}_{\pm0.29}$ | $61.94_{\pm0.23}$ |
| TabUnite(psk)-DDPM | $\mathbf{0.508}_{\pm0.013}$ | $98.91_{\pm0.10}$ | $97.41_{\pm0.3}$ | $99.38_{\pm0.32}$ | $\mathbf{67.36}_{\pm2.42}$ |
| TabFlow | $0.574_{\pm0.010}$ | $96.21_{\pm0.22}$ | $93.81_{\pm0.4}$ | $93.66_{\pm2.11}$ | $58.97_{\pm0.47}$ |
| oheFlow | $0.765_{\pm0.016}$ | $85.14_{\pm0.25}$ | $74.77_{\pm0.8}$ | $85.93_{\pm1.29}$ | $22.58_{\pm1.69}$ |
| TabUnite(i2b)-Flow | $0.544_{\pm0.007}$ | $97.79_{\pm0.14}$ | $96.43_{\pm0.3}$ | $98.08_{\pm0.79}$ | $60.76_{\pm0.29}$ |
| TabUnite(dic)-Flow | $0.561_{\pm0.013}$ | $98.03_{\pm0.29}$ | $96.44_{\pm0.3}$ | $97.71_{\pm1.03}$ | $60.63_{\pm0.59}$ |
| TabUnite(psk)-Flow | $0.534_{\pm0.006}$ | $98.36_{\pm0.21}$ | $96.48_{\pm0.3}$ | $98.16_{\pm0.72}$ | $62.65_{\pm0.58}$ |

| Methods | Default | | | | |
|---|---|---|---|---|---|
| | AUC ↑ | CDE ↑ | PCC ↑ | $\alpha$ ↑ | $\beta$ ↑ |
| TabDDPM | $0.752_{\pm0.008}$ | $98.20_{\pm0.13}$ | $97.16_{\pm0.6}$ | $96.78_{\pm0.44}$ | $\mathbf{53.73}_{\pm0.36}$ |
| oheDDPM | $0.557_{\pm0.052}$ | $51.39_{\pm2.72}$ | $51.11_{\pm1.7}$ | $5.26_{\pm0.17}$ | $0.16_{\pm0.12}$ |
| TabUnite(i2b)-DDPM | $0.762_{\pm0.005}$ | $\mathbf{99.00}_{\pm0.09}$ | $\mathbf{98.44}_{\pm0.2}$ | $\mathbf{99.12}_{\pm0.30}$ | $47.20_{\pm0.38}$ |
| TabUnite(dic)-DDPM | $0.763_{\pm0.007}$ | $98.24_{\pm0.19}$ | $92.65_{\pm1.6}$ | $96.97_{\pm0.51}$ | $50.57_{\pm0.36}$ |
| TabUnite(psk)-DDPM | $\mathbf{0.764}_{\pm0.005}$ | $98.65_{\pm0.07}$ | $92.87_{\pm2.0}$ | $98.27_{\pm0.20}$ | $50.86_{\pm0.28}$ |
| TabFlow | $0.742_{\pm0.008}$ | $96.42_{\pm0.27}$ | $93.27_{\pm1.0}$ | $93.15_{\pm1.82}$ | $53.25_{\pm0.74}$ |
| oheFlow | $0.759_{\pm0.005}$ | $92.78_{\pm0.33}$ | $86.75_{\pm1.4}$ | $93.20_{\pm0.93}$ | $31.09_{\pm0.63}$ |
| TabUnite(i2b)-Flow | $\mathbf{0.764}_{\pm0.004}$ | $98.04_{\pm0.26}$ | $96.88_{\pm0.9}$ | $98.29_{\pm0.48}$ | $48.47_{\pm0.40}$ |
| TabUnite(dic)-Flow | $0.759_{\pm0.007}$ | $97.36_{\pm0.38}$ | $90.84_{\pm1.8}$ | $96.04_{\pm1.25}$ | $51.02_{\pm0.42}$ |
| TabUnite(psk)-Flow | $0.763_{\pm0.006}$ | $97.85_{\pm0.54}$ | $94.51_{\pm2.1}$ | $97.80_{\pm1.58}$ | $49.64_{\pm0.56}$ |

| Methods | Magic | | | | |
|---|---|---|---|---|---|
| | AUC ↑ | CDE ↑ | PCC ↑ | $\alpha$ ↑ | $\beta$ ↑ |
| TabDDPM | $0.940_{\pm0.002}$ | $99.09_{\pm0.07}$ | $97.81_{\pm0.4}$ | $98.40_{\pm0.67}$ | $53.28_{\pm0.53}$ |
| oheDDPM | $0.940_{\pm0.002}$ | $\mathbf{99.23}_{\pm0.06}$ | $98.16_{\pm0.6}$ | $\mathbf{99.33}_{\pm0.27}$ | $58.64_{\pm0.44}$ |
| TabUnite(i2b)-DDPM | $\mathbf{0.944}_{\pm0.006}$ | $98.97_{\pm0.10}$ | $\mathbf{98.57}_{\pm0.5}$ | $97.54_{\pm0.44}$ | $65.92_{\pm0.34}$ |
| TabUnite(dic)-DDPM | $0.943_{\pm0.003}$ | $99.17_{\pm0.13}$ | $98.06_{\pm0.6}$ | $99.05_{\pm0.43}$ | $65.33_{\pm0.34}$ |
| TabUnite(psk)-DDPM | $0.939_{\pm0.003}$ | $99.04_{\pm0.09}$ | $98.03_{\pm0.7}$ | $98.41_{\pm0.47}$ | $58.37_{\pm1.42}$ |
| TabFlow | $0.938_{\pm0.002}$ | $97.74_{\pm0.40}$ | $97.20_{\pm0.8}$ | $93.23_{\pm1.02}$ | $68.17_{\pm0.41}$ |
| oheFlow | $0.930_{\pm0.003}$ | $96.77_{\pm0.73}$ | $97.30_{\pm0.5}$ | $92.48_{\pm1.54}$ | $50.56_{\pm0.69}$ |
| TabUnite(i2b)-Flow | $0.941_{\pm0.001}$ | $98.10_{\pm0.57}$ | $97.84_{\pm0.8}$ | $96.05_{\pm1.48}$ | $\mathbf{68.42}_{\pm0.50}$ |
| TabUnite(dic)-Flow | $0.940_{\pm0.002}$ | $98.39_{\pm0.14}$ | $98.03_{\pm0.5}$ | $96.54_{\pm0.42}$ | $68.07_{\pm0.50}$ |
| TabUnite(psk)-Flow | $0.940_{\pm0.002}$ | $98.12_{\pm0.37}$ | $97.83_{\pm0.8}$ | $95.69_{\pm1.54}$ | $67.40_{\pm0.61}$ |

| Methods | Shoppers | | | | |
|---|---|---|---|---|---|
| | AUC ↑ | CDE ↑ | PCC ↑ | $\alpha$ ↑ | $\beta$ ↑ |
| TabDDPM | $0.904_{\pm0.004}$ | $97.58_{\pm0.12}$ | $96.55_{\pm0.1}$ | $90.28_{\pm0.55}$ | $\mathbf{72.46}_{\pm0.51}$ |
| oheDDPM | $0.799_{\pm0.126}$ | $83.17_{\pm11.77}$ | $81.62_{\pm8.4}$ | $67.78_{\pm32.72}$ | $27.37_{\pm15.03}$ |
| TabUnite(i2b)-DDPM | $\mathbf{0.919}_{\pm0.010}$ | $\mathbf{98.77}_{\pm0.12}$ | $\mathbf{98.22}_{\pm0.1}$ | $\mathbf{97.81}_{\pm0.48}$ | $50.85_{\pm0.39}$ |
| TabUnite(dic)-DDPM | $0.910_{\pm0.005}$ | $97.82_{\pm0.11}$ | $96.14_{\pm0.3}$ | $95.50_{\pm0.53}$ | $55.34_{\pm0.61}$ |
| TabUnite(psk)-DDPM | $0.912_{\pm0.006}$ | $97.91_{\pm0.09}$ | $96.88_{\pm0.2}$ | $93.42_{\pm0.48}$ | $67.03_{\pm0.44}$ |
| TabFlow | $0.914_{\pm0.005}$ | $95.73_{\pm0.20}$ | $93.75_{\pm0.2}$ | $80.85_{\pm1.25}$ | $62.26_{\pm1.01}$ |
| oheFlow | $0.910_{\pm0.006}$ | $92.44_{\pm0.16}$ | $90.58_{\pm0.4}$ | $80.79_{\pm2.88}$ | $47.63_{\pm0.71}$ |
| TabUnite(i2b)-Flow | $0.916_{\pm0.005}$ | $97.56_{\pm0.08}$ | $96.82_{\pm0.4}$ | $96.71_{\pm1.67}$ | $55.51_{\pm0.88}$ |
| TabUnite(dic)-Flow | $0.903_{\pm0.006}$ | $96.72_{\pm0.10}$ | $94.95_{\pm0.3}$ | $95.09_{\pm0.71}$ | $52.64_{\pm0.57}$ |
| TabUnite(psk)-Flow | $0.911_{\pm0.007}$ | $97.61_{\pm0.10}$ | $96.69_{\pm0.3}$ | $95.40_{\pm1.04}$ | $56.41_{\pm0.58}$ |

| Methods | Bank | | | | |
|---|---|---|---|---|---|
| | AUC ↑ | CDE ↑ | PCC ↑ | $\alpha$ ↑ | $\beta$ ↑ |
| TabDDPM | $0.922_{\pm 0.002}$ | $98.40_{\pm 0.07}$ | $97.21_{\pm 0.3}$ | $92.00_{\pm 0.55}$ | $\mathbf{56.95}_{\pm 0.36}$ |
| oheDDPM | $0.881_{\pm 0.007}$ | $97.32_{\pm 0.35}$ | $94.55_{\pm 0.1}$ | $92.78_{\pm 1.00}$ | $35.70_{\pm 0.45}$ |
| TabUnite(i2b)-DDPM | $0.920_{\pm 0.005}$ | $\mathbf{99.45}_{\pm 0.08}$ | $\mathbf{98.69}_{\pm 0.2}$ | $99.02_{\pm 0.55}$ | $46.75_{\pm 0.31}$ |
| TabUnite(dic)-DDPM | $\mathbf{0.923}_{\pm 0.002}$ | $99.21_{\pm 0.04}$ | $98.46_{\pm 0.1}$ | $98.80_{\pm 0.37}$ | $47.68_{\pm 0.21}$ |
| TabUnite(psk)-DDPM | $0.918_{\pm 0.002}$ | $99.41_{\pm 0.04}$ | $98.71_{\pm 0.1}$ | $99.15_{\pm 0.32}$ | $45.45_{\pm 0.28}$ |
| TabFlow | $0.910_{\pm 0.003}$ | $96.70_{\pm 0.33}$ | $94.83_{\pm 0.5}$ | $84.74_{\pm 2.70}$ | $52.16_{\pm 0.41}$ |
| oheFlow | $0.902_{\pm 0.002}$ | $95.08_{\pm 0.22}$ | $93.37_{\pm 0.3}$ | $89.84_{\pm 1.56}$ | $40.51_{\pm 0.47}$ |
| TabUnite(i2b)-Flow | $0.918_{\pm 0.002}$ | $98.60_{\pm 0.21}$ | $97.75_{\pm 0.2}$ | $98.69_{\pm 0.77}$ | $48.39_{\pm 0.34}$ |
| TabUnite(dic)-Flow | $0.919_{\pm 0.002}$ | $98.40_{\pm 0.12}$ | $97.25_{\pm 0.4}$ | $\mathbf{99.49}_{\pm 0.27}$ | $48.32_{\pm 0.44}$ |
| TabUnite(psk)-Flow | $0.918_{\pm 0.002}$ | $98.88_{\pm 0.07}$ | $98.07_{\pm 0.2}$ | $98.48_{\pm 0.42}$ | $47.09_{\pm 0.26}$ |

| Methods | Cardio | | | | |
|---|---|---|---|---|---|
| | AUC ↑ | CDE ↑ | PCC ↑ | $\alpha$ ↑ | $\beta$ ↑ |
| TabDDPM | $0.801_{\pm 0.002}$ | $99.39_{\pm 0.02}$ | $96.99_{\pm 1.6}$ | $97.79_{\pm 0.19}$ | $50.20_{\pm 0.15}$ |
| oheDDPM | $0.800_{\pm 0.001}$ | $99.28_{\pm 0.10}$ | $98.91_{\pm 0.2}$ | $98.12_{\pm 0.45}$ | $49.59_{\pm 0.17}$ |
| TabUnite(i2b)-DDPM | $\mathbf{0.802}_{\pm 0.006}$ | $\mathbf{99.72}_{\pm 0.16}$ | $\mathbf{99.38}_{\pm 0.6}$ | $\mathbf{99.79}_{\pm 1.07}$ | $49.59_{\pm 0.61}$ |
| TabUnite(dic)-DDPM | $\mathbf{0.802}_{\pm 0.001}$ | $99.70_{\pm 0.02}$ | $97.33_{\pm 1.4}$ | $99.69_{\pm 0.12}$ | $49.49_{\pm 0.17}$ |
| TabUnite(psk)-DDPM | $\mathbf{0.802}_{\pm 0.001}$ | $99.69_{\pm 0.03}$ | $99.16_{\pm 0.4}$ | $99.68_{\pm 0.11}$ | $49.22_{\pm 0.21}$ |
| TabFlow | $0.794_{\pm 0.003}$ | $98.11_{\pm 0.11}$ | $93.64_{\pm 2.9}$ | $91.75_{\pm 0.56}$ | $51.94_{\pm 0.16}$ |
| oheFlow | $0.794_{\pm 0.002}$ | $94.31_{\pm 0.18}$ | $93.14_{\pm 0.4}$ | $79.61_{\pm 0.98}$ | $\mathbf{52.06}_{\pm 0.23}$ |
| TabUnite(i2b)-Flow | $0.801_{\pm 0.001}$ | $99.07_{\pm 0.05}$ | $93.46_{\pm 3.3}$ | $98.72_{\pm 0.69}$ | $49.76_{\pm 0.34}$ |
| TabUnite(dic)-Flow | $\mathbf{0.802}_{\pm 0.001}$ | $98.93_{\pm 0.18}$ | $95.06_{\pm 2.9}$ | $98.10_{\pm 0.71}$ | $50.05_{\pm 0.26}$ |
| TabUnite(psk)-Flow | $\mathbf{0.802}_{\pm 0.001}$ | $99.05_{\pm 0.10}$ | $95.66_{\pm 2.1}$ | $99.58_{\pm 0.13}$ | $49.23_{\pm 0.16}$ |

| Methods | Stroke | | | | |
|---|---|---|---|---|---|
| | AUC ↑ | CDE ↑ | PCC ↑ | $\alpha$ ↑ | $\beta$ ↑ |
| TabDDPM | $0.842_{\pm 0.035}$ | $\mathbf{99.10}_{\pm 0.09}$ | $93.98_{\pm 1.3}$ | $99.32_{\pm 0.58}$ | $\mathbf{72.94}_{\pm 0.65}$ |
| oheDDPM | $0.800_{\pm 0.036}$ | $98.93_{\pm 0.25}$ | $\mathbf{97.97}_{\pm 0.9}$ | $98.14_{\pm 0.55}$ | $56.36_{\pm 0.48}$ |
| TabUnite(i2b)-DDPM | $0.847_{\pm 0.038}$ | $98.99_{\pm 0.41}$ | $96.02_{\pm 1.5}$ | $99.43_{\pm 0.30}$ | $63.32_{\pm 0.53}$ |
| TabUnite(dic)-DDPM | $0.824_{\pm 0.021}$ | $\mathbf{99.10}_{\pm 0.12}$ | $97.56_{\pm 1.0}$ | $98.78_{\pm 0.50}$ | $64.01_{\pm 0.50}$ |
| TabUnite(psk)-DDPM | $0.856_{\pm 0.018}$ | $99.09_{\pm 0.14}$ | $97.43_{\pm 1.4}$ | $\mathbf{99.46}_{\pm 0.51}$ | $57.58_{\pm 0.62}$ |
| TabFlow | $\mathbf{0.868}_{\pm 0.035}$ | $97.79_{\pm 0.20}$ | $96.23_{\pm 1.9}$ | $94.43_{\pm 0.97}$ | $68.54_{\pm 0.95}$ |
| oheFlow | $0.854_{\pm 0.025}$ | $94.21_{\pm 0.81}$ | $90.86_{\pm 1.2}$ | $79.86_{\pm 2.01}$ | $52.91_{\pm 1.03}$ |
| TabUnite(i2b)-Flow | $0.840_{\pm 0.036}$ | $98.61_{\pm 0.20}$ | $97.47_{\pm 1.7}$ | $98.83_{\pm 1.15}$ | $64.97_{\pm 0.89}$ |
| TabUnite(dic)-Flow | $0.807_{\pm 0.019}$ | $98.43_{\pm 0.16}$ | $97.50_{\pm 2.1}$ | $98.72_{\pm 1.04}$ | $67.07_{\pm 0.57}$ |
| TabUnite(psk)-Flow | $0.857_{\pm 0.038}$ | $98.32_{\pm 0.09}$ | $96.36_{\pm 0.9}$ | $98.16_{\pm 0.73}$ | $62.84_{\pm 0.68}$ |

### D.3 GROUND TRUTH ASSESSMENT WITH SYNTHETIC TOY EXAMPLES

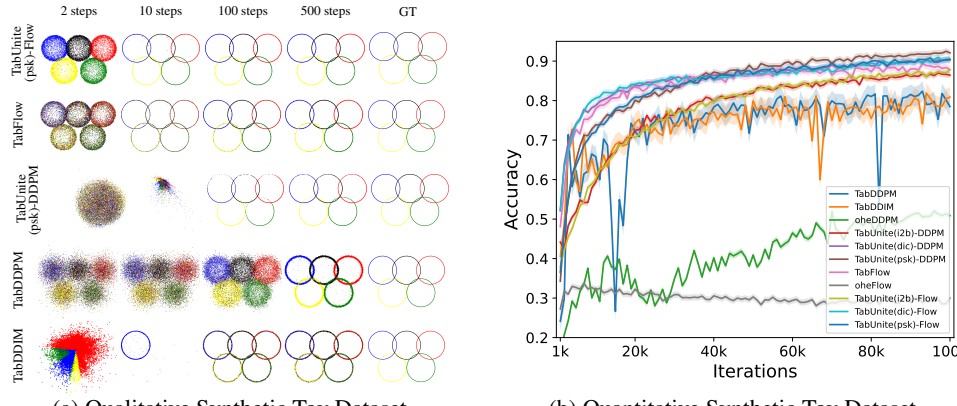

(a) Qualitative Synthetic Toy Dataset.  (b) Quantitative Synthetic Toy Dataset.

Figure 5: (a) The $x$-axis illustrates the sampling steps and the "Ground Truth" (GT) of the dataset whereas the $y$-axis depicts the methods. TabUnite methods are faithful in generating high-quality samples that match the ground truth in a short period of sampling duration. (b) The $x$-axis illustrates the training iterations whereas the $y$-axis depicts the accuracy of the generated categorical columns. Training TabUnite methods are stable and converge to a higher accuracy than TabDDPM.

Figure 5 illustrates a ground truth assessment of faithfulness using synthetic toy Examples. Details regarding how the dataset is curated can be found in Appendix C.1. A complete figure with all methods can be found in Figure 6.

**Qualitative Results.** We further demonstrate the effectiveness of our method in identifying ground truth relevance for data synthesis. We created a synthetic "Olympic" tabular dataset and visualized it qualitatively in terms of its structure (shape and sharpness of Olympic rings) and color. Our goal is to illustrate the integrity of our encoding method and sampling speed by mimicking the qualitative ground truth attributes of the real dataset. Our primary predefined model for comparison is TabDDPM. We also introduce TabFlow, a replica of TabDDPM except that we replace DDPM/Multinomial Diffusion with Flow Matching/Discrete Flow Models (Campbell et al., 2024).

Figure 5a displays the synthesized samples for TabUnite(psk)-Flow, TabFlow, TabUnite(psk)-DDPM, TabDDPM, and TabDDIM (Song et al., 2022b) across various sampling steps. As early as 10 steps, TabUnite(psk)-Flow converges, achieving high-quality structure and color in relation to the ideal "Ground Truth" (GT) visualization. However, there is no apparent "Olympic" structure for TabUnite(psk)-DDPM, TabDDPM, and TabDDIM. Although TabFlow presents an "Olympic" structure, it is difficult to identify the colors. TabFlow requires approximately 100 steps to differentiate between the colors clearly. Even at 500 steps, TabDDIM is underperforming in terms of its color whereas TabDDPM is lacking in its structure where the rings are visually less precise when compared to the GT. Therefore, the experiment highlights TabUnite(psk)-Flow's faithfulness and integrity in generating high-quality samples that match the ground truth in a short period of sampling duration.

**Quantitative Results.** In addition to our qualitative results, we further demonstrate quantitatively that our methods are faithful to the model's decision-making process by creating an additional synthetic toy dataset. In this dataset, categorical columns are created through an injective mapping from the numerical columns. We evaluate the synthesis of these categorical variables by taking the absolute value of the difference between the real value and the synthesized value. Our result in Figure 5b depicts the accuracy of the generated categorical columns over the number of training iterations. The models' accuracy ranking from high to low at 100k training iteration is as follows: TabUnite(psk)-DDPM, TabUnite(dic)-Flow, TabUnite(psk)-Flow, TabUnite(dic)-DDPM, TabFlow, TabUnite(i2b)-Flow, TabUnite(i2b)-DDPM, TabDDIM, TabDDPM, oheDDPM, oheFlow. It illustrates that training TabUnite models is stable and converges at a higher accuracy compared to their counterparts. Furthermore, the one-hot methods collapse, achieving very poor generalizations.

GROUND TRUTH ASSESSMENT ADDITIONAL COMPLETE RESULTS

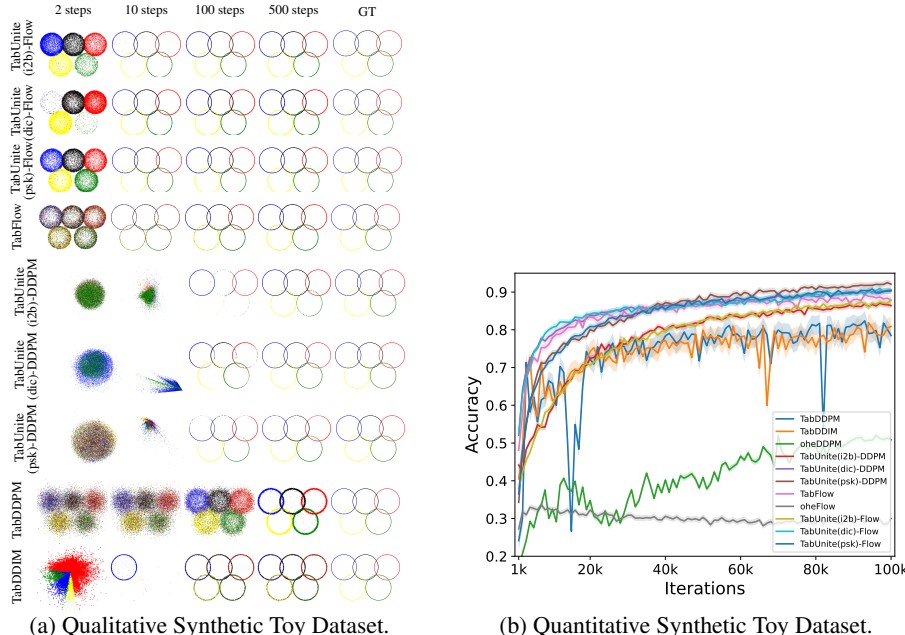

(a) Qualitative Synthetic Toy Dataset.  (b) Quantitative Synthetic Toy Dataset.

Figure 6: (a) The $x$-axis illustrates the sampling steps and the "Ground Truth" (GT) of the dataset whereas the $y$-axis depicts the methods. TabUnite methods are faithful in generating high-quality samples that match the ground truth in a short period of sampling duration. (b) The $x$-axis illustrates the training iterations whereas the $y$-axis depicts the accuracy of the generated categorical columns. Training TabUnite methods are stable and converge at a higher accuracy than TabDDPM.

### D.4 LOW-ORDER STATISTICS: COLUMN-WISE DENSITY ESTIMATION AND PAIR-WISE COLUMN CORRELATION

For reference, we also include supplementary results on the column-wise densities and pair-wise correlations achieved by baseline methods. The results are provided by TabSYN (Zhang et al., 2023).

The results for can be found in Tables 11 and 12.

Table 11: Comparison of column-wise density estimation scores (%). Values bolded in red is the best performing model for each dataset.

| Method | Adult | Default | Shoppers | Magic | Beijing | News |
|---|---|---|---|---|---|---|
| SMOTE | $98.40_{\pm 0.23}$ | $98.52_{\pm 0.15}$ | $97.32_{\pm 0.19}$ | $99.09_{\pm 0.05}$ | $98.15_{\pm 0.21}$ | $94.69_{\pm 0.46}$ |
| CTGAN | $83.16_{\pm 0.03}$ | $83.17_{\pm 0.04}$ | $78.85_{\pm 0.10}$ | $90.19_{\pm 0.08}$ | $78.61_{\pm 0.05}$ | $83.91_{\pm 0.02}$ |
| TVAE | $85.78_{\pm 0.08}$ | $89.83_{\pm 0.05}$ | $75.49_{\pm 0.06}$ | $91.75_{\pm 0.06}$ | $80.84_{\pm 0.06}$ | $83.38_{\pm 0.03}$ |
| GOGGLE | $83.03$ | $82.98$ | $77.67$ | $\mathbf{98.10}_{\pm 0.00}$ | $83.07$ | $74.68$ |
| GReaT | $87.88_{\pm 0.04}$ | $80.06_{\pm 0.06}$ | $85.49_{\pm 0.12}$ | $83.84_{\pm 0.09}$ | $91.75_{\pm 0.12}$ | OOM |
| STaSy | $88.71_{\pm 0.06}$ | $94.23_{\pm 0.06}$ | $90.63_{\pm 0.09}$ | $93.71_{\pm 0.13}$ | $93.29_{\pm 0.03}$ | $93.11_{\pm 0.03}$ |
| CoDi | $78.62_{\pm 0.06}$ | $84.23_{\pm 0.07}$ | $68.16_{\pm 0.05}$ | $88.44_{\pm 0.26}$ | $83.06_{\pm 0.02}$ | $67.73_{\pm 0.04}$ |
| TabSYN | $\mathbf{96.04}_{\pm 0.08}$ | $\mathbf{97.10}_{\pm 0.04}$ | $\mathbf{97.44}_{\pm 0.07}$ | $97.35_{\pm 0.12}$ | $\mathbf{97.76}_{\pm 0.04}$ | $\mathbf{94.26}_{\pm 0.05}$ |

Table 12: Comparison of pair-wise column correlation scores (%). Values bolded in red is the best performing model for each dataset.

| Method | Adult | Default | Shoppers | Magic | Beijing | News |
|---|---|---|---|---|---|---|
| SMOTE | $96.72_{\pm 0.29}$ | $91.59_{\pm 0.38}$ | $96.44_{\pm 0.22}$ | $96.84_{\pm 0.41}$ | $97.61_{\pm 0.35}$ | $94.62_{\pm 0.76}$ |
| CTGAN | $79.77_{\pm 1.20}$ | $73.05_{\pm 0.93}$ | $86.92_{\pm 0.16}$ | $93.00_{\pm 0.19}$ | $77.05_{\pm 0.08}$ | $94.63_{\pm 0.05}$ |
| TVAE | $85.85_{\pm 0.88}$ | $80.50_{\pm 0.95}$ | $81.33_{\pm 0.38}$ | $94.18_{\pm 0.49}$ | $81.99_{\pm 0.08}$ | $93.83_{\pm 0.09}$ |
| GOGGLE | $54.71$ | $78.06$ | $76.10$ | $90.53$ | $54.06$ | $76.81$ |
| GReaT | $82.41_{\pm 0.22}$ | $29.98_{\pm 0.12}$ | $54.84_{\pm 0.18}$ | $89.77_{\pm 0.40}$ | $40.40_{\pm 0.55}$ | OOM |
| STaSy | $85.49_{\pm 0.25}$ | $\mathbf{94.04}_{\pm 0.26}$ | $91.51_{\pm 0.15}$ | $93.39_{\pm 0.53}$ | $92.00_{\pm 0.10}$ | $\mathbf{96.93}_{\pm 0.04}$ |
| CoDi | $77.51_{\pm 0.08}$ | $31.59_{\pm 0.05}$ | $82.22_{\pm 0.11}$ | $93.47_{\pm 0.25}$ | $92.93_{\pm 0.15}$ | $88.90_{\pm 0.01}$ |
| TabSYN | $\mathbf{93.36}_{\pm 0.15}$ | $87.56_{\pm 1.02}$ | $\mathbf{93.55}_{\pm 0.08}$ | $\mathbf{96.81}_{\pm 0.12}$ | $\mathbf{94.20}_{\pm 0.13}$ | $95.84_{\pm 0.03}$ |

## D.5 HIGH-ORDER METRICS: $\alpha$-PRECISION AND $\beta$-RECALL

For reference, we also include supplementary results on the alpha-precision and beta-recall values achieved by baseline methods. The results are provided by TabSYN (Zhang et al., 2023).

The results for can be found in Tables 13 and 14.

Note that similar to the results obtained in TabSYN's paper, TabDDPM also collapses on the News dataset in our experiments.

Table 13: Comparison of $\alpha$-Precision scores. Higher values indicate superior results. Values bolded in red is the best performing model for each dataset.

| Methods | Adult | Default | Shoppers | Magic | Beijing | News |
|---------|-------|---------|----------|-------|---------|------|
| CTGAN | $77.74_{\pm 0.15}$ | $62.08_{\pm 0.08}$ | $76.97_{\pm 0.39}$ | $86.90_{\pm 0.22}$ | $96.27_{\pm 0.14}$ | $96.96_{\pm 0.17}$ |
| TVAE | $98.17_{\pm 0.17}$ | $85.57_{\pm 0.34}$ | $58.19_{\pm 0.26}$ | $86.19_{\pm 0.48}$ | $97.20_{\pm 0.10}$ | $86.41_{\pm 0.17}$ |
| GOGGLE | $50.68$ | $68.89$ | $86.95$ | $90.88$ | $88.81$ | $86.41$ |
| GReaT | $55.79_{\pm 0.03}$ | $85.90_{\pm 0.17}$ | $78.88_{\pm 0.13}$ | $85.46_{\pm 0.54}$ | $98.32_{\pm 0.22}$ | - |
| STaSy | $82.87_{\pm 0.26}$ | $90.48_{\pm 0.11}$ | $89.65_{\pm 0.25}$ | $86.56_{\pm 0.19}$ | $89.16_{\pm 0.12}$ | $94.76_{\pm 0.33}$ |
| CoDi | $77.58_{\pm 0.45}$ | $82.38_{\pm 0.15}$ | $94.95_{\pm 0.35}$ | $85.01_{\pm 0.36}$ | $98.13_{\pm 0.38}$ | $87.15_{\pm 0.12}$ |
| TabDDPM | $94.79_{\pm 0.27}$ | $98.27_{\pm 0.34}$ | $98.33_{\pm 0.40}$ | $93.35_{\pm 0.53}$ | $0.01_{\pm 0.73}$ | $0.00_{\pm 0.00}$ |
| TabSYN | $\mathbf{98.51}_{\pm 0.31}$ | $\mathbf{98.73}_{\pm 0.20}$ | $\mathbf{98.80}_{\pm 0.36}$ | $\mathbf{98.01}_{\pm 0.30}$ | $\mathbf{97.30}_{\pm 0.30}$ | $\mathbf{97.98}_{\pm 0.08}$ |

Table 14: Comparison of $\beta$-Recall scores. Higher values indicate superior results. Values bolded in red is the best performing model for each dataset.

| Methods | Adult | Default | Shoppers | Magic | Beijing | News |
|---------|-------|---------|----------|-------|---------|------|
| CTGAN | $30.80_{\pm 0.20}$ | $18.22_{\pm 0.17}$ | $31.80_{\pm 0.350}$ | $11.75_{\pm 0.20}$ | $34.80_{\pm 0.10}$ | $24.97_{\pm 0.29}$ |
| TVAE | $38.87_{\pm 0.31}$ | $23.13_{\pm 0.11}$ | $19.78_{\pm 0.10}$ | $32.44_{\pm 0.35}$ | $28.45_{\pm 0.08}$ | $29.66_{\pm 0.21}$ |
| GOGGLE | $8.80$ | $14.38$ | $9.79$ | $9.88$ | $19.87$ | $2.03$ |
| GReaT | $49.12_{\pm 0.18}$ | $42.04_{\pm 0.19}$ | $44.90_{\pm 0.17}$ | $34.91_{\pm 0.28}$ | $43.34_{\pm 0.31}$ | - |
| STaSy | $29.21_{\pm 0.34}$ | $39.31_{\pm 0.39}$ | $37.24_{\pm 0.45}$ | $\mathbf{53.97}_{\pm 0.57}$ | $54.79_{\pm 0.18}$ | $\mathbf{39.42}_{\pm 0.32}$ |
| CoDi | $9.20_{\pm 0.15}$ | $19.94_{\pm 0.22}$ | $20.82_{\pm 0.23}$ | $50.56_{\pm 0.31}$ | $52.19_{\pm 0.12}$ | $34.40_{\pm 0.31}$ |
| TabDDPM | $\mathbf{50.74}_{\pm 0.37}$ | $\mathbf{46.90}_{\pm 0.35}$ | $\mathbf{53.32}_{\pm 0.52}$ | $46.26_{\pm 0.35}$ | $0.02_{\pm 0.68}$ | $0.00_{\pm 0.00}$ |
| TabSYN | $45.13_{\pm 0.23}$ | $44.30_{\pm 0.29}$ | $48.68_{\pm 0.57}$ | $45.28_{\pm 0.40}$ | $\mathbf{55.50}_{\pm 0.21}$ | $35.70_{\pm 0.18}$ |

### D.6    DETECTION METRIC: CLASSIFIER TWO-SAMPLE TEST (C2ST)

The results for our C2ST tests can be found in Table 15. The majority of TabUnite methods outperform the baselines.

Table 15: Comparison of C2ST scores. Higher values indicate superior results. Values bolded in **red** is the best performing model for each dataset.

| Method | Adult | Default | Shoppers | Magic | Beijing | News |
|---|---|---|---|---|---|---|
| SMOTE | 97.10 | 92.74 | 90.86 | 99.61 | 98.88 | 93.44 |
| CTGAN | 59.49 | 48.75 | 74.88 | 67.28 | 75.31 | 69.47 |
| TVAE | 63.15 | 65.47 | 29.62 | 77.06 | 86.59 | 40.76 |
| GOGGLE | 11.14 | 51.63 | 14.18 | 95.26 | 47.79 | 7.45 |
| GReaT | 53.76 | 47.10 | 42.85 | 43.26 | 68.93 | – |
| STaSy | 40.54 | 68.14 | 54.82 | 63.99 | 79.22 | 52.87 |
| CoDi | 20.77 | 45.95 | 27.84 | 72.06 | 71.77 | 2.01 |
| TabDDPM | 97.55 | 97.12 | 83.49 | 99.98 | 95.13 | 0.02 |
| TabSYN | 99.86 | 98.70 | 97.40 | 97.32 | 96.03 | **97.49** |
| TabUnite(i2b)-DDPM | 99.40 | 98.77 | **97.96** | 98.52 | 93.80 | 96.25 |
| TabUnite(dic)-DDPM | 96.08 | 97.47 | 91.37 | **100.00** | **99.43** | 95.98 |
| TabUnite(psk)-DDPM | **99.91** | **99.85** | 94.22 | 99.16 | 98.10 | 96.27 |
| TabUnite(i2b)-Flow | 94.52 | 86.26 | 90.00 | 95.77 | 90.42 | 87.12 |
| TabUnite(dic)-Flow | 88.08 | 92.77 | 89.39 | 93.38 | 93.68 | 88.22 |
| TabUnite(psk)-Flow | 95.31 | 87.02 | 92.36 | 96.75 | 92.26 | 88.29 |

## D.7 PRIVACY METRIC: DISTANCE TO CLOSEST RECORD

The results of our DCR tests can be found in Table 16. As observed, we remain competitive but do not outperform TabSYN as the best method under this metric. This aligns with our hypothesis where TabSYN leverages a latent space thus, resulting in a lossy compression, improving their DCR scores.

Table 16: Comparison of DCR. Results closer to 50% indicate better performance on the test. Values bolded in **red** is the best performing model for each dataset.

| Methods | Adult | Default | Shoppers | Magic | Beijing | News |
|---|---|---|---|---|---|---|
| TabDDPM | $81.92_{\pm 0.13}$ | $64.05_{\pm 0.18}$ | $91.49_{\pm 0.07}$ | $63.51_{\pm 0.47}$ | $82.44_{\pm 0.09}$ | $59.09_{\pm 0.16}$ |
| TabSYN | $\mathbf{51.67}_{\pm 0.35}$ | $\mathbf{50.87}_{\pm 0.17}$ | $\mathbf{52.05}_{\pm 0.88}$ | $\mathbf{52.10}_{\pm 0.39}$ | $\mathbf{51.55}_{\pm 0.38}$ | $\mathbf{50.72}_{\pm 0.25}$ |
| TabUnite(i2b)-DDPM | $66.98_{\pm 0.45}$ | $90.50_{\pm 0.23}$ | $90.54_{\pm 0.65}$ | $95.12_{\pm 0.22}$ | $90.64_{\pm 0.49}$ | $90.19_{\pm 0.38}$ |
| TabUnite(dic)-DDPM | $71.10_{\pm 0.25}$ | $90.75_{\pm 0.44}$ | $93.15_{\pm 0.45}$ | $95.37_{\pm 0.17}$ | $91.99_{\pm 0.53}$ | $90.46_{\pm 0.26}$ |
| TabUnite(psk)-DDPM | $68.36_{\pm 0.22}$ | $90.34_{\pm 0.38}$ | $91.04_{\pm 0.55}$ | $95.38_{\pm 0.23}$ | $91.49_{\pm 0.69}$ | $90.40_{\pm 0.14}$ |
| TabUnite(i2b)-Flow | $53.87_{\pm 0.27}$ | $52.96_{\pm 0.44}$ | $59.66_{\pm 0.54}$ | $83.71_{\pm 0.28}$ | $54.33_{\pm 0.65}$ | $55.81_{\pm 0.11}$ |
| TabUnite(dic)-Flow | $65.35_{\pm 0.04}$ | $57.79_{\pm 0.26}$ | $72.16_{\pm 0.65}$ | $82.90_{\pm 0.46}$ | $60.97_{\pm 0.25}$ | $55.76_{\pm 0.51}$ |
| TabUnite(psk)-Flow | $68.30_{\pm 0.11}$ | $90.65_{\pm 0.23}$ | $91.96_{\pm 0.47}$ | $95.30_{\pm 0.44}$ | $91.47_{\pm 0.33}$ | $90.66_{\pm 0.41}$ |

## D.8 PREDICTION TASKS WITH TABUNITE ENCODING: RESULTS ON CLASSIFICATION AND REGRESSION TASKS

The following table illustrates our results on classification and regression tasks using the Adult, Stroke, and Cardio datasets. In this table, we use XGBoost to show that our encoding schemes work for predictive methods too. We trained and evaluated XGBoost using 5 different seeds.

Table 17: Comparison of XGBoost methods across different datasets. Higher AUC and lower RMSE indicate better performance. Values bolded in **red** is the best performing model for each dataset.

| Methods | Adult | | Stroke | Cardio |
|---|---|---|---|---|
| | Classification (AUC) | Regression (RMSE) | Classification (AUC) | Classification (AUC) |
| oheXGBoost | $\mathbf{0.913}_{\pm 0.0004}$ | $0.380_{\pm 0.0030}$ | $0.747_{\pm 0.0091}$ | $0.744_{\pm 0.0048}$ |
| i2bXGBoost | $0.908_{\pm 0.0007}$ | $0.389_{\pm 0.0026}$ | $0.759_{\pm 0.0078}$ | $0.765_{\pm 0.0006}$ |
| dicXGBoost | $0.908_{\pm 0.0003}$ | $\mathbf{0.379}_{\pm 0.0022}$ | $0.712_{\pm 0.0089}$ | $0.762_{\pm 0.0008}$ |
| pskXGBoost | $\mathbf{0.913}_{\pm 0.0010}$ | $0.383_{\pm 0.0026}$ | $\mathbf{0.811}_{\pm 0.0034}$ | $\mathbf{0.768}_{\pm 0.0016}$ |

Although the improvements are not as significant as generation, TabUnite encoding schemes still outperform one-hot in 3/4 of the datasets and attain the same best result as one-hot in 1/4 of the datasets.

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
