# OpenReview forum: "TabUnite: Efficient Encoding Schemes for Flow and Diffusion Tabular Generative Models"
_ICLR.cc/2025/Conference — ICLR 2025 Conference Withdrawn Submission_

### Official Review · Reviewer_VMCF · 2024-10-29

**Soundness:** 2
**Presentation:** 3
**Contribution:** 1
**Rating:** 3
**Confidence:** 4

**Summary:**

This paper proposes three methods for encoding categorical data in tabular data generation: Phase-Shift Keying, Dictionary, and Analog Bits. Additionally, it presents experimental results for various combinations using DDPM (Denoising Diffusion Probabilistic Models) and FlowMatching models as generative models. The results demonstrate that TabUnite achieves the best performance in tabular data generation.

**Strengths:**

This paper demonstrates several notable strengths:

1. Effective use of figures: The authors have utilized figures to enhance readers' understanding of the proposed methods. Visual representations can significantly aid in comprehending complex concepts, making the paper more accessible to a wider audience.

2. Comprehensive literature review: The paper provides a well-organized overview of related research in tabular data generation. This thorough review helps contextualize the study's contributions and demonstrates the authors' deep understanding of the field.

3. Extensive experimental validation: The researchers conducted a wide range of experiments with various combinations of encoding methods and generative models. This comprehensive approach effectively showcases the performance of the proposed TabUnite method.

**Weaknesses:**

This paper has some weaknesses as follows:

1. Lack of novelty: The paper essentially combines proposed encoding methods with diffusion-based generative models, which are known for their strong generative capabilities. This combination, while potentially effective, does not represent a significant novel contribution to the field.

2. Insufficient analysis: The most crucial aspect of this paper is the encoding part. However, the analysis falls short of demonstrating the individual impact of each encoding method. While the overall model performance is important, the paper would have benefited from a more detailed examination of how each encoding method (Phase-Shift Keying, Dictionary, and Analog Bits) performs independently.

**Questions:**

I'm curious about whether the proposed encoding methods are also effective in other baseline models.

Additionally, I'm interested in understanding how each encoding method affects the sampling results.

---

### Official Review · Reviewer_zvKe · 2024-10-31

**Soundness:** 3
**Presentation:** 4
**Contribution:** 2
**Rating:** 5
**Confidence:** 3

**Summary:**

This submission addresses generative modelling on tabular data. It highlights the challenges for generative modelling when dealing with continuous and categorical data inputs. It proposes two encoding schemes for such data to be used as pre-processing of the input to unify continuous and categorical inputs into a single representation.

These two encodings schemas are: **PSK Encoding**, which encodes categorical values using a circular, phase-based representation. and **Dictionary Encoding**, which encodes categorical values using a look-up embedding table function. The underlying generative models are diffusion (DDPM) and flow matching. The submission provides empirical results on a large set of real-world and synthetic datasets and performance of the proposed encoding as used with DDPM and flow matching is compared against multiple other generative models.

The main contribution of this paper (among the ones outlined by the authors) are the two proposed encoding schemes (PKS, Dic) and their comprehensive evaluation on (i) multiple dataset and against a third SOTA encoding schema (i2b) and (ii) against non diffusion / non flow matching generative models. Although this work indeed provides interesting insights, the results do not indicate a strong outperformance of the proposed encodings schemes as compared to the SOTA encoding i2b (including results from appendices on Bank, Cardio, Stroke datasets).

Further, looking at the proposed encoding schemes, I am wondering why they have to be static and are not learned via an embedding as done in e.g., [1] and [2]. I would love to see a comparison of the proposed static encoding schemes against the learned embeddings proposed from these related works.

[1] FinDiff: Diffusion Models for Financial Tabular Data Generation
Timur Sattarov, Marco Schreyer, Damian Borth
ACM ICAIF, 2023

[2] TabDiff: a Multi-Modal Diffusion Model for Multi-Modal Tabular Data Generation
Juntong Shi, Minkai Xu, Harper Hua, Hengrui Zhang, Stefano Ermon, Jure Leskovec
https://arxiv.org/abs/2410.20626, 2024

**Strengths:**

- **(S1):** this work presents new encoding schemes for categorical input variables in tabular data. The problem of unifying continuous and categorical input in tabular data is not solved and renders an interesting challenge for generative modelling on such data. I can see many important and useful applications once this challenge can be considered solved.

- **(S2):** this paper provides a comprehensive evaluation for the domain of generative modelling on tabular data including great insights about how different generative models perform on tabular data. Also, thanks for the additional details provided in the appendix.

- **(S3):** this paper is well written and easy to follow.

**Weaknesses:**

- **(W1)**: in my opinion, the main weakness of this work is the reported performance of the proposed encoding schemes as compared to the SOTA encoding scheme i2b. For some datasets the proposed encodings work better but provide only a marginal outperformance. For some other dataset SOTA encoding (i2b) outperform the proposed encoding schemes.

- **(W2)**: the second weakness is the missing comparison to learned embeddings as the ones presented in [1] and [2]. Comparing the proposed encoding scheme to learned ones would support the paper's comprehensive evaluation and be helpful for future research in this domain.

- **(W3)**: both proposed encoding schemes are rather fixed or hand-crafted in their nature and therefore limited in their potential use for encoding categorical input values. For example, **PSK encoding** is interesting and seems to work very well for periodic categorical values such as Mon, Tue, ect,...where the data points projected on the circular plane preserver the relationship i.e., last day of the week is next to first day of the week. However, it will not work for many other categorical values such as, e.g., city names where the relationship between data points can not be expressed through a continuous circular representation, but rather a geographical one or a learned representation. Similarly, holds true for **Dictionary Encoding**. This encoding does indeed preserves the intrinsic ordering among ordinal categorical data in the embedding space but again does not provide a solution to nominal categorical values such as cities (or others such as hierarchical structures represented as e.g., business unit codes), which could be represented through a learned embedding. Since this paper claims to be unifying in bringing continuous and categorical inputs together, I consider this an important point to be made.

**Questions:**

- **(Q1)**: this paper outlines really nicely in the introduction the challenges generative modeling is facing when dealing with tabular data. In this context, I am wondering why the third listed challenge is so restrictive: "...(3) learned latent embedding which is parameter inefficient.". The paper provides an example mentioning one-hot-encoding in this context, which leads to sparse representations not suitable for generative modeling. I fully agree with this statement but would love to hear the author's opinion about learned representations, which potentially might be able to encapsulate rich semantics of categorical values in a continuous space.

- **(Q2)**: Minor: in the examples explaining the proposed encoding schemes, I would recommend not using numbers but rather categorical values -> instead of saying $x^{cat} \in$ {1,2,3,4} is encoded to { -1, -0.5, 0, 0.5, 1}  it would be helpful to say $x^{cat} \in ${Chicago,New_York,Paris,Tokyo} is encoded to {-1, -0.5, 0, 0.5, 1}.

---

### Official Review · Reviewer_eB3e · 2024-10-31

**Soundness:** 3
**Presentation:** 3
**Contribution:** 2
**Rating:** 3
**Confidence:** 5

**Summary:**

This paper presents TabUnite, an encoding framework for tabular flow and diffusion models. It addresses the challenge of feature heterogeneity in tabular data generation. TabUnite uses novel encoding schemes (PSK, Dictionary, Analog Bits) to convert categorical to continuous features, enabling a single generative model to capture feature interrelationships. It combines these with FM and DDPM to create TabUnite models, which outperform baselines. A large-scale dataset is also curated to validate the importance of good encoding schemes.

**Strengths:**

- Originality: The paper introduces new encoding schemes (PSK, Dictionary, Analog Bits) for categorical features in tabular data, differing from traditional one-hot encoding. It approaches feature heterogeneity in tabular data generation with novel preprocessing strategies.

- Quality: The paper conducts extensive experiments on diverse datasets and against many baselines, and curates a large-scale heterogeneous dataset for comprehensive evaluation.

- Clarity: Clearly explains encoding schemes and their integration with generative models. The paper has a logical structure guiding readers through the research process.

- Significance: TabUnite models achieve better performance, useful for industries relying on tabular data. Advances tabular data generation research, inspiring future work on encoding techniques.

**Weaknesses:**

- Lack of Intuitive Comparison: While the paper describes the proposed encoding schemes (PSK, Dictionary, Analog Bits) in detail, it could provide a more intuitive comparison with network-based auto-encoding. For example, a straightforward baseline should be TabSyn with the latent diffusion kept the same as DDPM/Flow in TabUnite, and compare the results. Besides, TabSyn is missing in Tab 3.

- Novelty: I think the paper only concentrates on data processing, and in my point of view the novelty is limited considering ML conference. Especially, diffusion/flow based methods have been well studied in this domain.

**Questions:**

See weakness.

---

### Official Review · Reviewer_5Shu · 2024-11-04

**Soundness:** 2
**Presentation:** 4
**Contribution:** 2
**Rating:** 3
**Confidence:** 4

**Summary:**

The paper presents a unified approach for generating synthetic tabular data by addressing the challenges of heterogeneous data representation (i.e., both continuous and categorical variables). TABUNITE proposes three encoding schemes—PSK Encoding, Dictionary Encoding, and Analog Bits—to convert categorical features into continuous representations. This unified approach allows for the application of Flow Matching and Diffusion-based models across all feature types. The authors demonstrate that these encoding schemes significantly enhance model performance in tabular data generation tasks and achieve competitive results across various benchmarks.

**Strengths:**

1. $\textbf{Innovative Handling of Categorical Variables}$: The paper addresses one of the most critical and currently popular challenges in tabular representation—handling categorical variables. It introduces three distinct encoding methods and demonstrates their effectiveness, contributing valuable insights to the field.

2. $\textbf{Clear and Readable Presentation}$: The paper is well-structured and written in a clear, accessible manner, making it easy to follow and understand the proposed methods.

**Weaknesses:**

1. $\textbf{Excessive Emphasis on Novelty}$: The paper presents three new methods for processing categorical variables and applies them to existing models, claiming this as the foundation of a new model. However, this contribution appears overstated, as the primary novelty lies in the preprocessing techniques rather than in the development of a fundamentally new model. Comparatively, the impact of these additions may be limited.


2. $\textbf{Overemphasis on the Novelty of Dictionary Encoding}$: The Dictionary Encoding approach essentially applies a min-max scaling to categorical variables, assigning them continuous values within a fixed range. While this is useful for handling ordinal data, the technique is quite similar to standard scaling methods and may not represent a significant innovation. Consequently, the novelty of the Dictionary Encoding approach may be somewhat overstated in the context of its actual contribution.

3. $\textbf{Inadequate Handling of Categorical Variables in Analog Bits Encoding}$: The Analog Bits encoding method represents categorical variables as continuous binary vectors. However, this approach may not fully capture the discrete nature of categorical data, as it implies distances between categories that don’t inherently exist. This can blur the boundaries between distinct categories, potentially reducing the model’s ability to differentiate them effectively—especially in tasks where clear categorical distinctions are important. This issue is further compounded by the decoding process, which interprets the model’s continuous outputs through a simple thresholding method, limiting precision in recreating categorical identities.

**Questions:**

Have you explored whether applying the proposed preprocessing methods to existing tabular data synthesis models could enhance their performance? It would be interesting to see if these encoding schemes provide similar benefits when integrated with other generative models.

---

### Official Review · Reviewer_TQab · 2024-11-05

**Soundness:** 3
**Presentation:** 3
**Contribution:** 2
**Rating:** 5
**Confidence:** 4

**Summary:**

This paper presents TabUnite, an novel framework that applies efficient encoding schemes for tabular data in flow and diffusion-based generative models. Recognizing the challenges posed by heterogeneous feature types in tabular data, TabUnite utilizes three encoding methods—PSK, Dictionary, and Analog Bits—to convert categorical features into compact continuous representations. This unified continuous representation allows for the application of a single generative model across both continuous and categorical features, enhancing modeling of complex inter-feature relationships. Extensive experiments demonstrate that TabUnite achieves competitive or superior performance on several benchmarks compared to state-of-the-art models.

**Strengths:**

1.	The paper is well-organized and easy to follow. It provides a thorough background on diffusion model / flow matching and encoding schemes. Additionally, the authors offer a detailed description on PSK Encoding, Dictionary Encoding, and Analog Bits, and the generative models training.
2.	The authors provide comprehensive experiments with multiple baselines on tabular data generation, covering extensive datasets.
3.	The proposed encoding schemes are simple yet effective. The space complexity also provided for detailed analysis.

**Weaknesses:**

1.	The authors proposed three straightforward yet effective encoding schemes, but the design choices and analysis behind them require clarification. For instance, in phase-shift keying (PSK) encoding, why is a uniformly distributed phase used? For categorical attributes like nationality, distances between categories may vary, so uniform phase distribution might not be ideal. Why is the encoding structured as in Eq. (1), and could a learnable phase bias parameter improve it? Further explanation on the design choices for each encoding scheme would enhance understanding.
2.	While the proposed schemes yield improved performance, the intuition and underlying reasons for this improvement are unclear. It would be beneficial to offer more justification and explanation for these schemes, especially in comparison to one-hot encoding. The authors mention that generative models are susceptible to underfitting in high-dimensional spaces, but no supporting evidence is provided in the paper.
3.	The proposed encoding schemes appear versatile and could potentially benefit various tasks, such as classification and regression. Are these schemes more effective than one-hot encoding for predictive tasks? Including results from prediction tasks would help demonstrate the broader applicability of these schemes.

My current score is low, but I am open to raising it if the authors can adequately address my concerns.

**Questions:**

Please see the weakness part.

---

### Note · Authors · 2024-11-12

I have read and agree with the venue's withdrawal policy on behalf of myself and my co-authors.